# Multidimensional social influence drives leadership and composition-dependent success in octopus–fish hunting groups

Eduardo Sampaio ●[1,2,3,4] ✉, Vivek H. Sridhar[2,3,4,5], Fritz A. Francisco ●[6,7], Máté Nagy ●[2,3,4,8,9], Ada Sacchi[2], Ariana Strandburg-Peshkin ●[3,4,5], Paul Nührenberg[2,3,4], Rui Rosa[1,10], Iain D. Couzin ●[2,3,4,11] & Simon Gingins[2,3,4,11]

Collective behaviour, social interactions and leadership in animal groups are often driven by individual differences. However, most studies focus on same-species groups, in which individual variation is relatively low. Multispecies groups, however, entail interactions among highly divergent phenotypes, ranging from simple exploitative actions to complex coordinated networks. Here we studied hunting groups of otherwise-solitary *Octopus cyanea* and multiple fish species, to unravel hidden mechanisms of leadership and associated dynamics in functional nature and complexity, when divergence is maximized. Using three-dimensional field-based tracking and field experiments, we found that these groups exhibit complex functional dynamics and composition-dependent properties. Social influence is hierarchically distributed over multiscale dimensions representing role specializations: fish (particularly goatfish) drive environmental exploration, deciding where, while the octopus decides if, and when, the group moves. Thus, 'classical leadership' can be insufficient to describe complex heterogeneous systems, in which leadership instead can be driven by both stimulating and inhibiting movement. Furthermore, group composition altered individual investment and collective action, triggering partner control mechanisms (that is, punching) and benefits for the de facto leader, the octopus. This seemingly non-social invertebrate flexibly adapts to heterospecific actions, showing hallmarks of social competence and cognition. These findings expand our current understanding of what leadership is and what sociality is.

Collective behaviour emerges from the network of interactions among individual parts. It is central to coordinated functioning across scales of organization, including physical[1], cellular[2] and social[3,4] systems. Heterogeneity, driven by genetic[4–6], physiological[6,7] and informational[4,5,8] differences among individuals, plays a vital role in explaining the functional complexity of collectives. We see the emergence of such complex behaviours in the division of labour among cells[2] or insect societies[3]. The same functional dynamics also enable several alternatives of decision-making in groups, which can be placed over a despotic–democratic axis, that is, one individual leader or shared leadership[9–11]. Despite this, groups in which differences among system components may be expected to be greatest—multispecies animal groups—have received comparatively little attention.

Common across terrestrial and aquatic ecosystems, multispecies groups are composed of individuals with species, and thus function-specific, characteristics linked to specialized strategies driven

**Fig. 1 | Multispecific hunting assemblages and overview of the core methodology. a**, Multispecific hunting groups can be composed of several species, such as the day octopus *O. cyanea*, long-barbel goatfish *P. macronemus*, yellow- and blue-phase goldsaddle goatfish *P. cyclostomus*, lyretail grouper *V. louti* and blacktip grouper *E. marginatus*. Short names are found below common names in the figure and are used hereafter (Supplementary Table 1). **b**, To reconstruct the hunting scenes in 3D, animals were tracked manually in each of the camera videos. We used a stereocamera rig (top) from which habitat features were identified (middle) and camera positions derived (in red) using structure from motion, enabling the calculation of relative 3D track positions (bottom). Colours show individuals present from different species, with large dots representing manual annotations and small dots showing smoothed interpolated tracks. **c**, Finally, we extracted pulls and anchors from our data using sets of minimum–maximum–minimum dyadic distances ($t_1$, $t_2$, $t_3$), also registering at $t_3$ if the initiation was successful, that is, recruited the follower (resulting in a pull, and the outcomes 'pulling' and 'follower') or unsuccessful, that is, the initiator moved back towards the partner it failed to engage in moving (resulting in an 'anchor', and the outcomes 'anchored' and 'anchoring'). Note that the verbal conjugation highlights the individual that has influence on the movement of the other. In further figure captions, statistical details are given in the respective supplementary tables.

by divergent evolutionary histories[12,13]. Yet, the members of the group as a whole may exhibit a shared objective, such as during collective hunting. In general, groups of animals hunting together can increase the likelihood of acquiring information[14] about prey locations[15,16], potentially leading to division of roles and consequent role specialization[17,18]. Such is also true in the case of interspecific interactions, in which differently evolved hunting strategies can more easily lead to role specialization, as seen particularly in coordinated pairwise associations[12,19]. Thus, multispecies groups offer a unique opportunity to quantify how networks of highly heterogeneous behavioural phenotypes achieve complex coordinated action.

Despite having diverged at the vertebrate–invertebrate division ~550 million years ago (Ma)[20], otherwise-solitary foraging octopuses can be accompanied by several fish with which they share a generalist diet (that is, feeding on smaller crustaceans, fish and molluscs)[21,22]. Octopuses typically forage by moving along the reef searching for hidden prey using their arms, either by probing into crevices or by fully enveloping corals or rocks in web-overs, a general strategy that has been termed 'speculative hunting'[23,24]. Accompanying fish

species possess different predation strategies that evolved according to their specific ecological niches, including active bottom-churning feeding, for example, long-barbel goatfish *Parupeneus macronemus* (hereafter 'barbel goatfish') and yellow and blue goldsaddle goatfish *Parupeneus cyclostomus* (hereafter 'yellow goatfish' and 'blue goatfish', respectively); stalking open water predation, for example, lyretail grouper *Variola louti* (hereafter 'lyretail'); and sit-and-wait ambush predation, for example, blacktip grouper *Epinephelus fasciatus* (hereafter 'blacktip') (Fig. 1a, Supplementary Table 1 and Supplementary Video 1).

Octopus–fish hunting groups have been mostly considered as 'nuclear–follower' (or 'producer–scrounger') systems[10,25], in which the octopus is the nucleus of the group as it stimulates and maintains group cohesion[26]. Indeed, fishes accompanying octopuses or other predators with similar foraging strategies (for example, moray eels) can increase prey capture success as they gain access to prey that is being flushed out by the nuclear predator[19,21,27]. Concurrently, changes in octopus predation success are unclear, with suggestions that the octopus might disregard[24], or is being exploited by, fish[28]. Thus, in these systems, one would expect despotic leadership regarding movement, with fish

exclusively following the octopus[21,24,28]. However, recent qualitative observations of octopuses following fish[22,27] suggest the existence of more complex group organizations, in which movement leadership can dynamically change among different individuals. Nonetheless, in the absence of quantitative evidence, the organizational properties and functional nature (that is, exploitative/competitive or collaborative) of the octopus–fish system have remained speculative.

Here we investigate the functional dynamics emerging within multispecies hunting groups composed of one octopus and individuals of several fish species. Using three-dimensional (3D) field-based tracking of individuals participating in octopus–fish hunting groups, we quantify social influence hierarchies as well as composition-driven emergent behaviours and group properties. Using two full-frame wide-lens cameras on an aluminium structure, we obtained 3D reconstructions of environmental features and camera positions using structure from motion, and calculated the 3D trajectories of the individuals comprising the hunting groups (Fig. 1b)[29]. After a combined total of ~120 h of diving, we successfully filmed and tracked 13 group hunting scenes composed of *Octopus cyanea* and multiple fish partners, divided into 107 subgroup blocks of 100s (see Methods, ref. 30 and Supplementary Video 2 for an example).

In a group, the movement of individuals can exert influence on others, offering valuable insights into leadership dynamics, decision-making processes and the emergence of specialized roles. Theoretically, individuals can exert influence through two (not necessarily mutually exclusive) routes: either by initiating movement that is followed by their group mates or by refusing to move and thereby inhibiting others from doing so. Movement initiation is usually linked to exploring new environments and finding locations where there may be prey[15,25]. Conversely, movement inhibition (that is, stopping movement by others) could shed light on other important facets of social influence, such as dominance, codependence or interdependence, between group members[31]. Moreover, within each type of influence, a meaningful duality, is the potential difference between the frequency of successful events and the likelihood of being successful per event (that is, the 'efficiency'). Frequency, that is, the number of successful initiation movements, can be translated into functional influence within the group and has been widely used in the field of animal collective behaviour as the principal metric of leadership (for example, refs. 10,11). On the other hand, efficiency in influence could be a measurement of 'initiator quality' ('leader quality'[32]), as it weighs the frequency of successfully influencing others with the effort spent in doing so.

Octopus–fish hunting groups exhibit punctuated motion, with abrupt changes in speed with respect to both individual and group motion (Supplementary Fig. 1 and Supplementary Video 2) that can be well characterized by a 'pull' and 'anchor' physical process[11]. This method gauges the relative movement between two individuals, in which one becomes an initiator (moving away) and the other is a potential follower. If an initiation event recruits a following individual, it is termed a 'pull', creating positive feedback in terms of motion. Conversely, there also exists a potential negative feedback mechanism via 'anchoring', whereby the potential follower does not do so and the initiator subsequently returns to, and is thus 'anchored' by, that individual. Thus, we use the following terminology: while in the former scenario individuals are 'pulling' a 'follower', in the second, we have individuals that are 'anchoring' and those that are 'anchored' (Fig. 1c and see also Methods for further explanations). Moreover, by adding both pulling and anchoring frequencies, we calculated each individual's general influence over the movement of others (whether by inducing or inhibiting it) and defined the most influential member as the group's de facto leader. Lastly, to provide a multiscale overview of influence, we also quantified both pulling and anchoring frequencies between a given focal individual and the centroid of the group (see Methods for details and other complementary analyses).

In addition to pulling and anchoring, we also quantified other meaningful aspects of individual and collective movement, including individuals' speed, tortuosity of movement (low tortuosity equating to highly directed motion), distance to the group centre (centroid), the relative angle between the heading direction of the initiator and follower, and the relative angle between the individual's heading direction and its direction to the group centroid, as well as group-level properties, such as the mean distance of individuals to the group's centroid (that is, the spread of the group) as well as kinematics of the group's centroid, including displacement (travelled distance), speed and tortuosity (details in Methods).

## Leadership in multispecies groups

Contrary to what would be expected from nuclear–follower dynamics, leadership—as classically defined by the frequency of successful movement initiations (that is, pulls)—in octopus–fish hunting groups is demonstrably not despotic, but shared (Fig. 2 and see Supplementary Fig. 2 and Tables 2–10 for statistical models and pairwise post hoc comparisons). However, individuals do not have egalitarian leadership status; species identity was highly relevant to determine initiator–follower roles during hunting, forming hierarchical networks of social influence (Wald test, $\chi^2 = 63.351$, $P < 0.0001$; Fig. 2). Functionally, goatfishes (particularly blue goatfish individuals) emerge as the main drivers of movement initiation, that is, their frequency of initiations and pulls per minute (Fig. 2a) being far greater than those of non-goatfish species, including the octopus (Tukey honest significant differences (HSD) with correction; all comparisons $P < 0.05$; Fig. 2c and Supplementary Tables 2 and 3). Similar results are found when analysing only the first puller of a given follower (that is, the first individual moving in pulls where multiple individuals subsequently move; Supplementary Fig. 3 and Supplementary Table 11), confirming that these fish are the main drivers of group movement. Moreover, the same pulling influence outcomes are found even when intraspecific interactions are removed (Supplementary Fig. 4 and Supplementary Table 12). Goatfish, particularly the blue and yellow goatfish (same species), are active predators with high mobility that find and corner prey together with other conspecifics[33]. These fish seem to use similar strategies in interspecific groups, serving as social cues for others. However, while blue goatfish have the highest pulling influence, equally mobile yellow goatfish (presumably a younger phase of the same species) exert less influence on the movement of others. This contrast between phases suggests changes in hunting performance, or potentially different strategies, with age[34]. Furthermore, we found markedly different network structures depending on assessing the frequency (Fig. 2a and Supplementary Table 2) or the efficiency (Fig. 2b and Supplementary Table 7) of being followed by others. This highlights the existence of several pathways of influence within groups, an issue that has been scarcely investigated to date[32], particularly using a quantitative approach with empirical data.

By exploring this different facet of social influence, we found that both octopus and blacktips are highly efficient at both pulling and anchoring (Tukey HSD, $P < 0.05$; Fig. 2b and Supplementary Tables 6 and 7), thus possessing what has been referred to as possible 'leader quality'[32]. We anticipated that blacktips, as they rely more on ambush predation, would exclusively follow others. This was not the case. Their unexpected status as highly efficient pullers and anchorers may be due to a higher signal-to-noise ratio in movement (following signal detection theory, as in ref. 35), comparatively to other species. As blacktips are ambush predators, they spend longer periods relatively immobile than other hunting partners (moving less than 1 cm s$^{-1}$; Tukey HSD, $P < 0.05$; Supplementary Table 13 and Supplementary Fig. 5). Therefore, initiation (or lack) of movement on their part may provide a clearer and more salient (greater 'signal to noise', using the terminology of signal detection theory) cue to other group members that prey may or may not be nearby, serving as 'quality' indicators. Nonetheless, octopuses were the main functional anchorers in multispecific

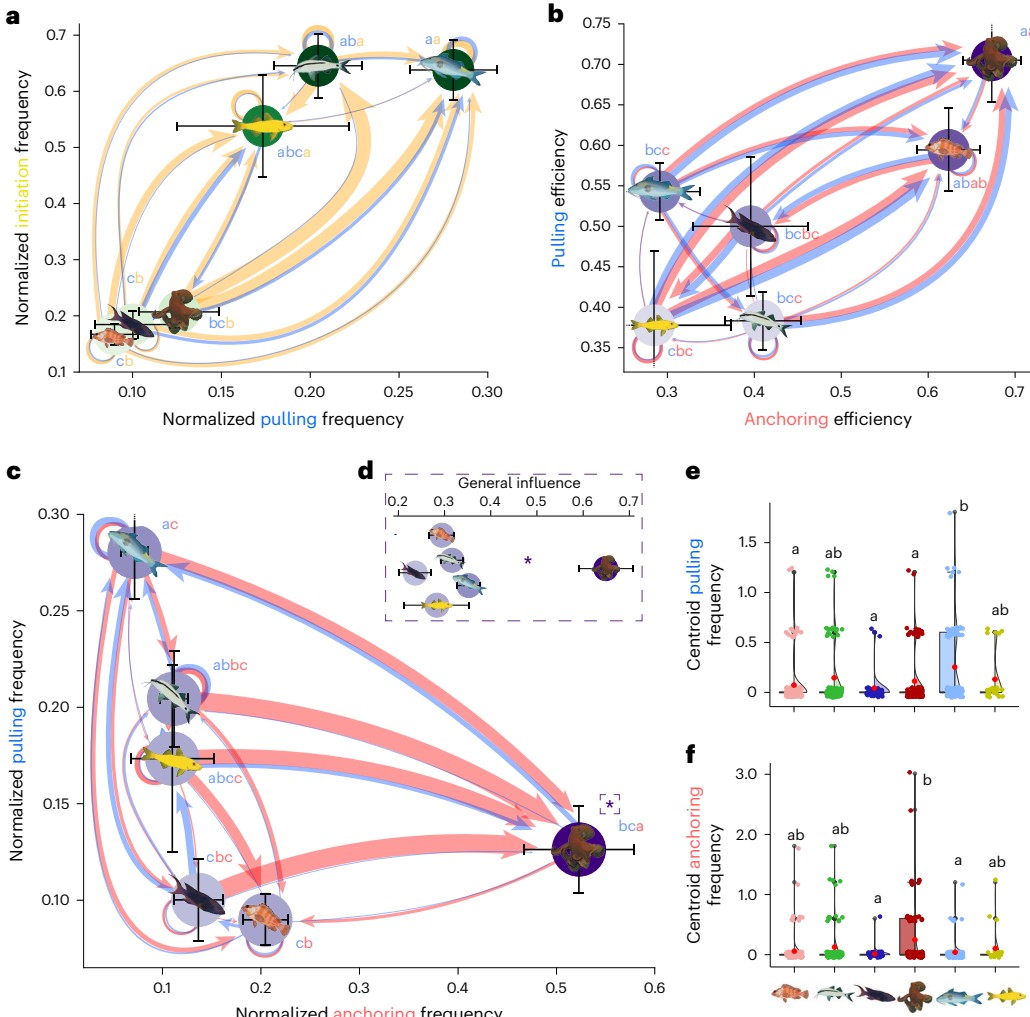

**Fig. 2 | Multidimensionality of leadership. a,b,** Directed hierarchical social networks based on species-specific normalized frequency of pulls (Supplementary Table 2) and initiations (Supplementary Table 4) (**a**) and pull efficiency (that is, probability of successfully leading, Supplementary Table 7) and anchoring efficiency (that is, probability of anchoring, Supplementary Table 8) (**b**), on the *x* and *y* axes, respectively. **c,** Functional influence that each species has in pulling (Supplementary Table 2) and anchoring (Supplementary Table 6) other individuals, represented by normalized frequency of both parameters. The purple asterisk refers to significant differences in general influence (calculated as the sum of pulling and anchoring frequencies), that is, the 'purple dimension' (the fusion between blue and red). **d,** The general functional influence that each species has in influencing the movement of others (Supplementary Table 14). Frequencies are normalized by the number of individuals present in the group in that given subgroup. Positions along the axes are calculated as the mean score ± s.e. across all species, and node colours are the mean score of both axes combined, with edge directions pointing towards the larger value in the dyad for each parameter and with width representing the differential of values. **e,f,** The influence at a group level that each species has via pulling (**e**) and anchoring (**f**) the group's centroid. Data points, violin plots and box plots are used. For box plots, boxes represent the 25% and 75% quartiles, with the centre (50%) being the median and red dots indicating the mean. Whiskers represent the equal or lower and upper values of 1.5 times the interquartile range (between 25% and 75%). To illustrate multiple-comparison statistics, different letters between species indicate significant differences and are coloured according to the corresponding parameter in each axis. For example, a species labelled 'a' is significantly different from species labelled 'b' and 'c'. However, a species labelled 'ab' is different only to species labelled 'c', and not 'a' and 'b'. As upper and lower standard error estimates are similar, error bars are truncated to fit graph limits where applicable (dashed lines).

groups, that is, frequency of events anchoring other individuals per minute (Tukey HSD, all *P* < 0.05; Fig. 2c and Supplementary Table 6), representing the main driver of movement inhibition in others. Such a prominent role in this dimension is probably linked to the strong dependence of other partners on the octopus' unique abilities to flush out otherwise-inaccessible prey with its flexible arms[21]. This strong dependence also explains why octopuses are the most efficient in pulling others. That is, if the octopus does not follow, fish tend to stop and turn back, but if the octopus moves to a new location (for example, following a fish initiator), (other) fish will follow to attempt capturing flushed out prey. Together with such marked efficiency, if we consider general functional influence on others'

movement, that is, frequency of events both initiating or inhibiting the movement of others, the octopus emerges as the most influential individual in the group, the de facto leader or 'decider' (Tukey HSD, *P* < 0.05; Fig. 2d and Supplementary Table 14).

By analysing pull–anchor dynamics and social influence at a group level (that is, not relatively to another individual, but relative to the group's centroid), we found a similar influence distribution, whereby blue goatfish are the main pullers (Tukey HSD, *P* < 0.05; Fig. 2e and Supplementary Table 15) and octopuses are the main anchorers of the group's centroid (Tukey HSD, *P* < 0.05; Fig. 2f and Supplementary Table 16). Thus, in general collective movement, group dynamics can be divided into two functional roles: goatfish, in particular blue goatfish,

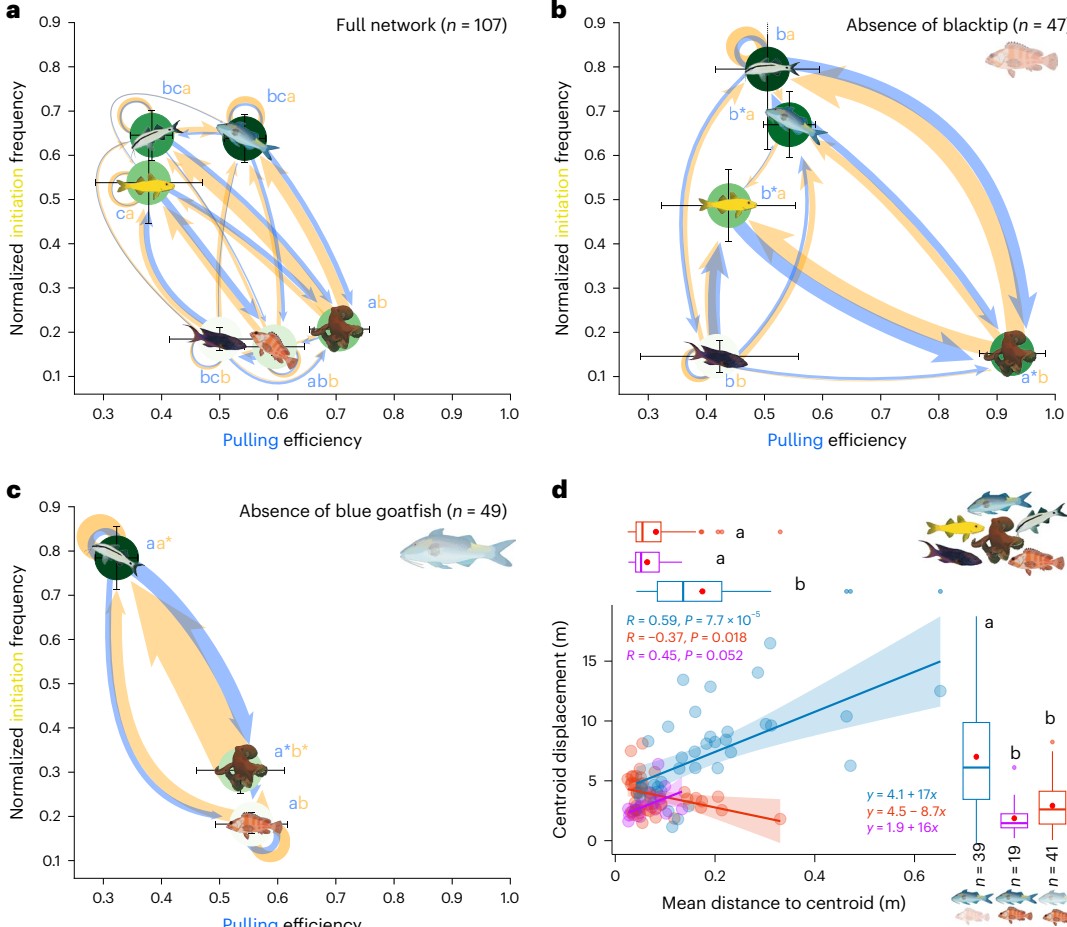

**Fig. 3 | Group composition effects on hierarchical networks, individual investment and group properties. a–c,** Using a full network using all species and database as a baseline (**a**), species composition effects, that is, absence of the main anchorers (blacktip, **b**) or absence of the main pullers (blue goatfish, **c**), significant alterations to nodes and network are highlighted based on species-specific mean ± s.e. normalized initiation frequency and pulling efficiency. Networks were created using a similar methodology to Fig. 2, adding asterisks representing significant differences compared with the opposite conditions: absence versus presence of blacktips in **b** (Supplementary Tables 17 and 18) and absence versus presence of blue goatfish in **c** (Supplementary Tables 21 and 23). Axes limits are maintained among **a–c** to better emphasize differences and are coloured according to the corresponding parameter in each axis. To illustrate multiple-comparison statistics between species, different letters indicate significant differences between species. **d,** Changes in group properties fundamental to habitat exploration depending on the combined factorial possibilities of extreme phenotypes present: blue goatfish, blacktips and both together (absence of both was not included given low sample size; Supplementary Tables 26 and 27). For box plots, boxes represent the 25% and 75% quartiles, with the centre (50%) being the median and red dots indicating the mean. Whiskers represent the equal or lower and upper values of 1.5 times the interquartile range (between 25% and 75%).

explore space and determine where the group may go, whereas the octopus decides if, and when, the group goes. This functional role division between initiators and a decider in octopus–fish groups is akin to what has been qualitatively described (yet to be quantitatively confirmed) in the field for kin-related groups of hamadryas baboons in the 1960s[36]. In this system, the suggested dynamics rely on lower-ranked initiators (young males) providing different direction options for older males to choose from, the latter acting as the deciders in the troop. If true, multispecies groups composed of differently specialized, unrelated, individuals can achieve the same functional dynamics and coordination shown by highly social species.

## Composition effects on group action and individual investment

To assess how extreme phenotypes impact individual and group-level properties, we conducted directed network and pull–anchor analyses on groups in which these phenotypes were not present (Fig. 3a–c). When blacktips—which can exhibit a relatively strong anchoring (Fig. 2)—were absent from the multispecies assemblage (Fig. 3b), the frequency of

initiations was maintained (Tukey HSD, $P > 0.05$ for species and group level; Supplementary Table 17), but there was an increase in the efficiency at which individuals could pull others, at both the individual and species levels (Tukey HSD, $P < 0.05$ for all species except lyretail, and $P < 0.001$ for the group level; Supplementary Table 18). In terms of group-level characteristics, these shifts manifested in increases in centroid travelled distance (or centroid displacement, $P < 0.0001$; Supplementary Table 19) and mean individual distance to the centroid of the group (or group spread, $P < 0.0001$; Supplementary Table 20), as well as a strong positive correlation between these two variables (Pearson correlation, $R = 0.6$, $P < 0.0001$), when blacktips are absent (Supplementary Fig. 6). Thus, blacktips function mainly as negative feedback for other group members, with their absence resulting in higher environment exploration.

As seen by their initiation frequency and pulling influence (Fig. 2), blue goatfish play a key role in both group mobility (considerably impacting movements of others) and structure. However, this species is not always present in these multispecies assemblages, raising questions as to how groups' influence dynamics respond when it is absent.

While groups with blue goatfish often also encompass the yellow phase of the same species and lyretails, it was still possible to analyse network and pull–anchor outcomes at the group level, and at a species-specific level, for those species that were present independently of the occurrence of blue goatfish. (1) Does another species take over its role? (2) Do all species compensate by increasing their pulling influence and initiation frequency? (3) Does the group fundamentally change and become more 'anchored'?

We found that when blue goatfish were absent (Fig. 3c), there existed not only an individual-level increase in initiations by individuals of the remaining species present (Tukey HSD, $z$-value = 4.421, $P < 0.0001$; Supplementary Table 21), but also a concurrent increase in the frequency of anchors (failures to initiate; $P < 0.0001$; Supplementary Table 22 and Supplementary Fig. 7) and a consequent declining pulling efficiency ($P = 0.0082$; Fig. 3c and Supplementary Table 23). This indicates that these individuals exhibited a higher movement initiation effort, yet with lower rates of success (compared with when the blue goatfish is present). Furthermore, we found that these increases in initiations and anchors (failures) were species specific (asterisks in Fig. 3c). Both octopus and barbel goatfish increased their initiations (both $P < 0.01$; Supplementary Table 21 and Supplementary Fig. 8) and failed more (both $P < 0.0001$; Supplementary Table 22 and Supplementary Fig. 8), with the octopus' pulling success in particular significantly decreasing ($P = 0.0124$; Supplementary Table 23 and Supplementary Fig. 8). There were no differences in these metrics for blacktips (Supplementary Tables 21–23). In other words, in the absence of the main functional puller species, while octopus and barbel goatfish increase their investment in searching for prey (and consequently in group movement) by increasing their initiations, blacktips do not.

These changes in pull–anchor metrics in the absence of the main pullers were associated with the group becoming more compact around the centroid ($P < 0.0001$; Supplementary Table 24 and Supplementary Fig. 10), but statistically not less mobile ($P > 0.05$; Supplementary Table 25 and Supplementary Fig. 9), although the positive correlation between these two group properties shifted to negative (Pearson correlation, $R = -0.36$, $P = 0.012$). However, by explicitly analysing groups according to the presence of extreme phenotypes—(1) with blue goatfish (blacktips absent), (2) with blacktips (blue goatfish absent) and (3) with both species (Fig. 3d)—we found that groups containing blacktips become less mobile and more compact around the centroid ($P < 0.05$ compared with blue-goatfish-only groups; Fig. 3d and Supplementary Tables 26 and 27). On the other hand, only in the absence of the blacktips' negative feedback did the positive feedback of blue goatfish via pulling result in increased group displacement and spread, increasing environment exploration (correlation slope: $y = 4.1 + 17x$; Pearson correlation, $R = 0.59$, $P < 0.0001$; Fig. 3d).

Consistent with this idea, analyses of individual-level kinematics showed that the initiator's speed, together with tortuosity, distance to centroid and angle between the direction of travel between the initiator and potential follower, is relevant in predicting the success of initiations (binomial generalized (non-)linear mixed models (GLMM), $n = 1,180$, all $P < 0.02$; Supplementary Table 28 and Supplementary Fig. 10). Moreover, species-level kinematics (Supplementary Tables 29–36) show that blue goatfish consistently exhibited higher speed than blacktips, both on average ($P < 0.0001$; Supplementary Table 29 and Supplementary Fig. 11) and during initiations ($P = 0.0063$; Supplementary Table 33 and Supplementary Fig. 12), hinting at how the presence of blue goatfish and absence of the blacktips result in enhanced positive feedback and consequent environmental exploration by the group. One consequence of these inherent movement differences is that it was not possible to disentangle between species identity and mean variations in kinematics, for example, speed, per se as the principal mechanism driving changes in environmental exploration. Nevertheless, our analyses enable us to state that, whether due to kinematics or species identity, these changes are tightly linked to the presence and absence of these extreme phenotypes.

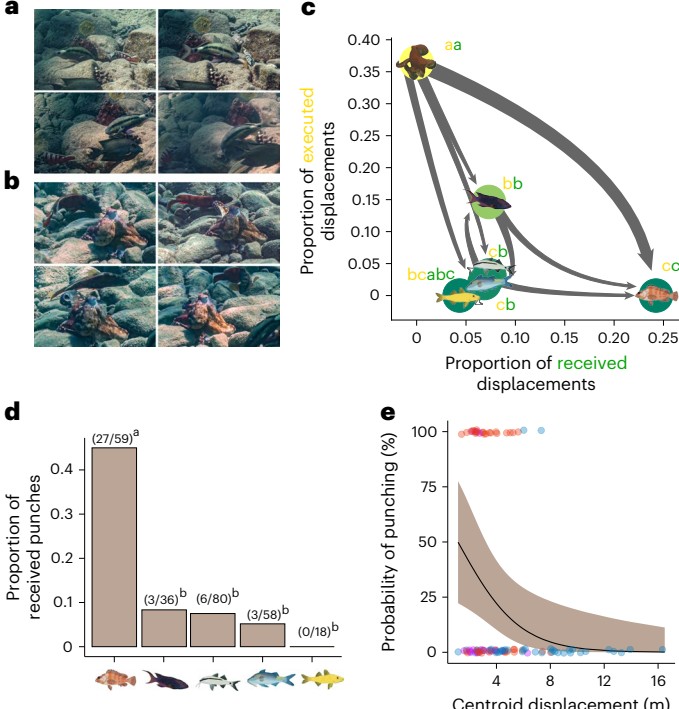

**Fig. 4 | Emergence of partner control mechanisms and its drivers. a**, A series of images from a video recording exemplifies an aggressive displacement from one fish to another, with the barbel goatfish darting towards a blacktip with its dorsal fin up. **b**, Another image series depicts a punch, that is, a direct negative feedback mechanism from the octopus towards a specific fish. **c**, The relative ranking of each species towards others in terms of aggressive interactions via received and executed displacements, using a similar methodology to Fig. 2 (Supplementary Table 37). Displacements are normalized by the number of groups and blocks where species are present, and are therefore given as a proportion. As the octopus is the most dominant individual, subsequent analyses are focused on recipients of the octopus' aggressive actions, that is, species that are punched by the octopus. **d**, Species-specific proportions and frequency of individuals targeted by octopus punches (Supplementary Table 38). **e**, Data points (occurrence or non-occurrence: 1 and 0, respectively) and estimated probability distribution (centre line and shaded areas indicate the mean ± 95% CI) of the octopus punching a given fish partner depending on subgroup displacement (Supplementary Table 42). Data points are coloured depending on the combined factorial possibilities of extreme phenotypes present: blue goatfish (blue), blacktips (red) and both together (purple). To illustrate multiple-comparison statistics, different letters between species indicate significant differences and are coloured according to the corresponding parameter in each axis.

## Direct partner control mechanisms

Large investment asymmetries within groups, together with negative group and individual-level impacts, can lead to the emergence of direct negative feedback between individuals through partner regulation mechanisms in the form of aggressive actions[37]. Thus, another dimension defining leadership and hierarchical control in groups is dominance, measured as the unequal ranking of individuals involved in aggressive actions, both as actors and receivers[11,38]. We found that in octopus–fish hunting groups, two forms of direct aggressive interactions exist: fish can displace others by darting towards them (Fig. 4a and Supplementary Video 3), and octopuses can displace fish by punching them (Fig. 4b and Supplementary Video 4). Punching involves an explosive motion of one arm directed at a specific hunting partner, which actively displaces it to outer areas of the group temporarily or permanently[22]. Across species, we found that the octopus was the main interspecific regulator of the group, perpetrating a disproportionate number of aggressive actions towards fish partners (proportion

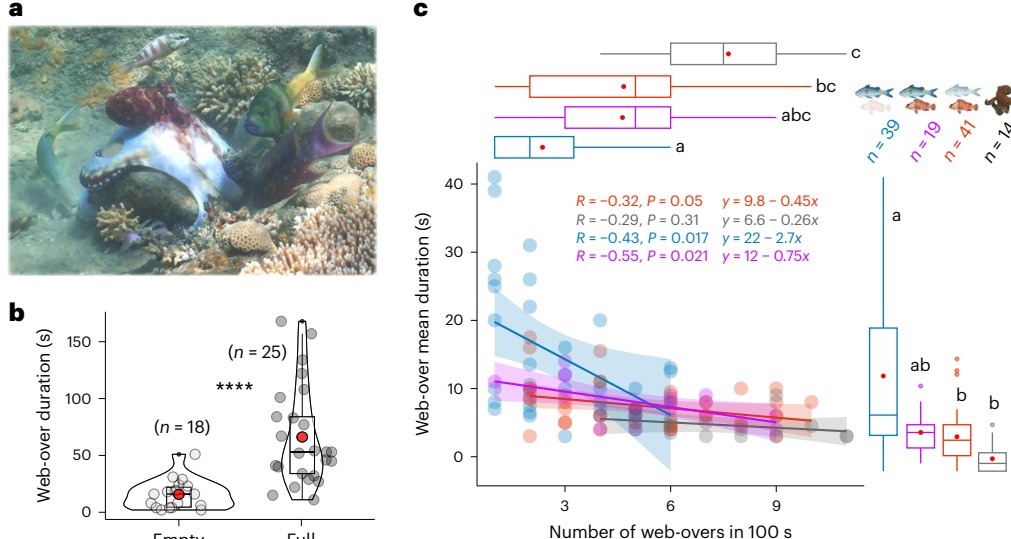

**Fig. 5 | Octopus foraging strategy and benefits. a,** An octopus web-over, a distinctive behavioural motif characterized by a whitening and conspicuous expansion of the interbrachial web skin over a specific habitat feature expressed when capturing prey within coral and rock crevices. **b,** Octopuses perform longer web-overs over structures with food than over empty ones, based on data from a field experiment. ****Significance level of $P < 0.001$ (Supplementary Table 44). **c,** Variations in octopus foraging strategy, that is, web-over duration (Supplementary

Table 45) and frequency (Supplementary Table 46), across the factorial possibilities of presence of extreme phenotypes (including hunting alone) based on the duration (success) and frequency (investment) of web-overs (Supplementary Tables 45 and 46). Pearson's $R$ and $P$ value are given in the figure. For box plots, boxes represent the 25% and 75% quartiles, with the centre (50%) being the median and the red dots indicating the mean. Whiskers represent the equal or lower and upper values of 1.5 times the interquartile range (between 25% and 75%).

tests with continuity corrections, all $\chi^2 \geq 6.016$, $P < 0.05$; Supplementary Table 37) while receiving none ($P < 0.05$, except yellow goatfish; Supplementary Table 37), thus emerging as the most dominant individual (Fig. 4c). Moreover, blacktips were the main target of regulation by other group members ($P < 0.05$, except for yellow goatfish; Fig. 4c and Supplementary Table 37), especially by the octopus ($P < 0.0001$; Fig. 4c and Supplementary Table 37; compared with other species, $P < 0.01$; Fig. 4d and Supplementary Table 38). As expected from sit-and-wait predators, while blacktips can be occasionally important in group movement decisions as 'quality indicators' (Fig. 2b), they perform very little initiations in complete assemblages and, unlike other species, refrain from increasing their initiation investment even when the main functional pullers are absent, thus mostly acting as opportunistic exploiters of other individuals' movement. Thus, in the presence of the blacktips, octopuses exhibit a much-increased propensity to engage in partner regulation ($n = 107$, $P = 0.0006$; Supplementary Table 39 and Supplementary Fig. 13a). However, punching is significantly decreased by the presence of blue goatfish ($P = 0.0011$; Supplementary Table 40 and Supplementary Fig. 13b), with a potentially larger effect than the blacktips' presence (Supplementary Table 41 and Supplementary Fig. 14), indicating that the presence of the main puller species helps stabilize social interactions. We hypothesized that such punching downregulation was related to increased exploration of the environment, which, from a functional perspective, can lead to finding more prey. Indeed, we found that the occurrence of punching is negatively correlated with the displacement of the group's centroid ($P = 0.0153$; Fig. 4e and Supplementary Table 42), indicating that this partner control mechanism acts to release the system from excessive negative feedback, resulting in enhanced environmental exploration and consequent prey-finding opportunities.

## Group composition effects drive the leader's benefits

To understand the functional consequences of these species-dependent modifications to the system's properties, we can ask how foraging strategies and benefits for the parties involved are impacted. As benefits

for fish associating with animals that flush prey (for example, octopus and moray eels) are already established in the literature[19,21,27,28], to understand the functional nature of these groups (that is, exploitation/ competition versus collaboration, parasitism/commensalism versus mutualism), we investigated the existence of possible benefits for octopuses in multispecies groups. We quantified the investment and temporal characteristics of the main and most conspicuous foraging-related behaviour exhibited by the octopus: its 'web-over' (Fig. 5a)[24,39]. As the octopus' morphology prevents direct measurement of prey intake or numbers, we used web-over characteristics as a proxy for prey capture attempts by octopuses, similar to how fast strikes are used as a proxy for prey captures by fish (for example, refs. 19,21). To experimentally validate the observational parameters and remove possible confounds, we performed a field experiment in which empty and food-baited (that is, full) structures were placed near the octopus (Supplementary Fig. 15). We found that octopuses were equally likely to perform web-overs on empty and full structures, but nearly always (around 95% of the times) performed them over structures that were previously attacked by fish ($n = 113$, $P = 0.004$; Supplementary Table 43, Supplementary Fig. 16 and Supplementary Video 5). Moreover, we found that web-over duration was independent of fish presence (Supplementary Table 44 and Supplementary Fig. 17), but that web-overs on full structures were longer than those on empty structures ($n = 43$, $P < 0.0001$; Fig. 5b and Supplementary Table 44). Thus, octopuses actively incorporate social information from fish when making prey-oriented decisions, and increases in web-over duration are tightly linked to successful food acquisition or prey capture, not, for example, potential kleptoparasitism avoidance.

Solitary foraging in octopuses is highly speculative[23,24], with large numbers of short-term web-overs occurring over habitat features while they move across the substrate (mean ± s.e., solitary foraging: 7.64 ± 0.40 web-overs in 100 s, lasting 4.61 ± 0.30 s; Supplementary Video 6). Octopus foraging when the blacktip was present remained highly speculative as when hunting alone, with web-overs still occurring at high frequency (4.67 ± 0.41 events, up to 10 events within 100 s) and with short duration (7.41 ± 0.31 s) (octopus–blacktip comparisons both $P > 0.05$; Fig. 5c and Supplementary Tables 45 and 46).

Thus, as expected, there are only marginal or non-existent gains for the octopus when hunting with blacktips. However, in the presence of blue goatfish (more markedly so with the simultaneous absence of blacktips), the octopus' foraging strategy became considerably more directed and efficient: web-overs occurred at less than a third of the frequency (2.36 ± 0.26 events), but lasted more than thrice as long (13.82 ± 1.18 s) when compared with hunting alone (octopus–blue goatfish comparisons both $P < 0.001$; Fig. 5c and Supplementary Video 7). In addition, the temporal characteristics of web-overs in these groups were also significantly different from those in groups with blacktips (blue goatfish comparisons, all $P < 0.05$; Fig. 5c), indicating increased success rates with less investment (correlation slope: $y = 22 - 2.7x$), representing a robust shift in octopus foraging strategy. This is associated with greatly increased group movement and spread, and thus increased environmental exploration, creating more prey-finding opportunities and acquisition for the octopus (and likely present fish partners), thus curtailing the need for partner control mechanisms from the de facto leader.

Combining the identified patterns of movement influence and foraging strategies reveals specialized hunting roles within the group: goatfish explore space, finding different prey locations, thereby providing options, while the octopus predominantly decides which option to take and attempts to capture the prey. Together, these joint actions can yield higher prey capture success rates for the octopus (this study) and for fish[21,27,28]. In terms of ecological functionality, for fish, the octopus serves as a specialist that they can guide to specific locations (see also refs. 19,27) to produce prey otherwise unattainable[21]. For the octopus, fish (in particular goatfish) act as an 'extended sensorial system' (sensu ref. 14) that samples larger spatial areas of the environment at a faster rate than the octopus could via direct sensing. This allows the octopus to filter the number of possible food sources via social information, seemingly saving energy (that is, less web-over effort to find prey) and moving only towards high-quality options (that is, containing prey). This invertebrate acts as the main controller and leader in the group, providing decisive feedback, both indirectly through its movement influence and directly through partner control mechanisms towards partners that invest less.

This functional complexity and its dynamic nature appear to distinguish this system from what is known of other interspecific hunting associations, such as badger–coyote[40] or mixed-species bird flock systems[12,41] (albeit these are understudied systems), as well as the well-studied pairwise moray eel–grouper system[19], in which direct partner control mechanisms seem absent. The exhibited range of partner-dependent behavioural flexibility, especially concerning the use of social information when deciding to switch foraging strategies and whom and when to punch, indicates that day octopuses have hallmarks of (heterospecific) social competence and cognition. Such ability to flexibly respond to non-predatory heterospecific actions within a complex dynamical system is rare (or under-reported) in animals. One particularly known example is cleaning stations. Here iterative interactions between resident clients and cleaner wrasses make clients rely on dynamic positive (cooperation) and negative (cheating) feedback mechanisms to choose among stations to visit, which conversely changes the behaviour of the cleaners themselves, leading to the emergence of a 'biological market'[42]. However, most animals that develop complex interactions with heterospecifics share complex social lives with conspecifics a priori, in principle transferring similar decision-making rules to heterospecific scenarios[43]. Day octopuses, often considered solitary or at least asocial with conspecifics, can have a sophisticated cross-species social life without relying on previous interactions with conspecifics. This, to our knowledge, is unique in invertebrates and extremely rare in the animal kingdom.

Our findings show how organisms phylogenetically separated for ~550 Ma can coordinate during collective hunting based on multidimensional and multiscale influence networks. Moreover, the underlying nature of such groups can vary gradually over an exploitative/collaborative (or parasitic/commensalistic versus mutualistic) axis, driven by changes in group properties, functionality and the emergence of direct feedback and control mechanisms dependent on species composition and behaviour. This hierarchically self-organized and complex system comprising highly divergent individuals provides conceptual and empirical evidence for the necessity of a new, multidimensional framework explaining leadership. Only by quantifying and integrating influence across multiple dimensions and scales, such as movement modulation, control and dominance, and ecological functionality, can we ascertain what truly drives social interactions, collective behaviour and leadership in animals.

## Methods

Procedures were approved by the Max Planck Institute of Animal Behaviour, the Department of Agriculture and Fisheries Ethics Committee from Queensland, Australia, and the Great Barrier Reef Marine Parks Authority under permit G23/47925.1.

### Location and recording procedures
Using SCUBA, we recorded interspecific hunting scenes between *O. cyanea* and multiple partners. Fieldwork spanned 1 month, between 1 October 2018 and 1 November 2018 (29.5577° N, 34.9519° E, Eilat, Israel), in a total of ~120 h of diving (~60 h each diver). Dives were performed two to three times per day, at relatively shallow depths (5–15 m), allowing ~2–3 h underwater per day, complying with local scientific diving regulations. Given that these hunts are not stationary, we adopted a search-and-follow procedure while maintaining a distance of >5 m to minimize disturbing natural interactions. We used two full-frame Sony Alpha 7SII with Zeiss 2/25 mm wide lenses mounted on an aluminium structure, as a stereocamera setup (hereafter 'stereocamera rig'; Fig. 1b). A third full-frame Sony Alpha 7SII with Sony f/4 24–70 mm lens served as a focal camera for the octopus (hereafter 'zoom camera'). Together with an additional ~30 min dataset in Egypt (El Quseir, 26.1014° N, 34.2803° E), video from the 'zoom camera' was used to quantify temporal characteristics of web-overs in solitary octopuses. We registered web-over frequency as the least subjective component of octopus foraging, and considered web-over duration as the timespan in which the octopus exhibits whitening of the interbrachial web skin over a specific habitat feature. There were no significant differences between data collected in Israel and Egypt during solitary hunting, regarding either web-over frequency (Poisson GLMM, $n = 14$, $z = -0.661$, $P = 0.509$) or duration (Gaussian GLMM, $n = 107$, $z = -1.166$, $P = 0.244$). Videos were all filmed at 25 fps with 4k resolution, and cameras were synchronized in Adobe Premiere via the timestamp of an underwater horn at the start of all recordings.

### Field experiments
Complementary field experiments were conducted at Lizard Island Research Station in Australia, where baited (prawn *Penaeus monodon*) or empty U-PVC tee-fitting structures were placed ~50 cm from octopuses, to gauge how web-over characteristics, that is, occurrence (interaction with the structure or not) and duration of web-overs, were impacted by (1) presence of food, (2) previous strikes on the structure by fish and (3) presence of fish (Supplementary Fig. 16 and Supplementary Video 5). We used 5 min (300 s) of web-over duration as maximum trial time, as this meant that the octopus had taken the structure back to its den (in these cases, web-over duration was not considered).

### Scene reconstructions and supplementary data sorting
Taking advantage of stereopsis provided by the overlap of the two cameras' field views, the stereocamera rig allowed reliable and accurate 3D tracking of overall group collective movement and 3D reconstruction of

habitat features. First, using computer-vision methods, videos were run through a structure-from-motion and multiview stereo open-source pipeline named 'colmap'[44]. The concept of structure from motion allows the retrieval of 3D information from two-dimensional (2D) images, by matching key points of a stationary background over several video frames. Adding two camera positions per time frame (multiview stereo) then allows key points to be triangulated in 3D space and decreases camera projection errors. In addition, 'colmap' also performs intrinsic camera calibration, undistorting images due to different lens types and camera features, and extrinsic calibration by calculating the position of one camera relative to the other per time frame. This extrinsic calibration per time frame yields the camera positions relative to the reconstructed habitat at all time frames, allowing us to recreate the path taken by the cameras while filming (see example in Fig. 1b), along with a high-resolution 3D spatial reconstruction in which all habitat features across hunting scenes are detailed. We manually tracked individuals in the videos using the software Computer Vision Annotation Tool. We annotated three frames per second, which yielded a time resolution of 0.33 s for animal movement. Combining both cameras views, this sampling effort represents a total of ~500,000 individual annotations. We annotated all individuals in a collective hunt, specifically the left eye of the octopus and the end of the rostrum of the fish, ensuring consistency between different annotators.

We then used another software developed to incorporate the previously tracked animals in each camera in the 'colmap' habitat models and camera paths, 'multiviewtracks' or 'mvt'[29]. Similarly to how key points were triangulated in the habitat reconstruction phase, individual positions were triangulated from the stereocamera rig's known camera relative positions, and their movements reconstructed in three dimensions from the entire camera path (and therefore nullifying camera motion). Next, we specified the known world distance between the two cameras for scale (1.2 m) and obtained individual trajectories in real $xyz$ coordinates. Taking advantage of knowing the real-world distance between cameras, we were also able to calculate reconstruction accuracy and reprojection camera errors. Reconstructions from 'mvt' had a remarkable median accuracy of 0.0001 m (that is, 0.1 mm) and a 3-sigma limit of 0.01 m (3 × s.d. error, that is, ~99.7% of the data). Finally, to further account for potential jittering arising from manual tracking, we searched and removed position outliers ($x$, $y$ or $z$ coordinate values diverging three times the standard deviation from the last three annotated frames), linearly interpolated missing values (up to 12 frames, or 4 s) and applied a Savitzky–Golay filter to smoothen data encompassing a time window of 19 frames (package 'SciPy').

In total, 3.5 h of collective hunting was reconstructed (example in Fig. 1b and Supplementary Video 2), composed of 13 different scenes representing different groups of interspecific hunting. From these 13 groups, we collected data and maintained the individual identity of 13 octopuses, 22 blacktips, 12 barbel goatfish, 10 lyretails, 20 blue goatfish and 4 yellow goatfish. The size of the groups varied between 2 and 10 individuals, with an average presence of 5.72 individuals, always with 1 octopus present. The shortest scene recording of a hunting group was 100 s, while the largest reached approximately 1,800 s (limited by camera battery). As movement frequency was nonlinear across time, particularly in long-duration groups (Supplementary Fig. 1), we standardized length variability to better show the dynamics of the groups. We used the smallest scene length of a hunting group (100 s) to divide all hunting groups into 100 s blocks, totalling 107 hunting subgroups. In the end, our reconstructed subgroups provided a sample size of 132 blacktips, 95 barbel goatfish, 42 lyretails, 107 octopuses, 102 blue goatfish and 23 yellow goatfish (see 'Statistics' for techniques used to deal with data dependency). We further re-ran analyses on the main estimated parameters using 200 s (Supplementary Fig. 18a,b) and 300 s (Supplementary Fig. 18c,d) blocks and confirmed that the specific time length chosen did not impact our results. Details on the number of blocks per group and overlap of species in each subgroup

or group are provided in ref. 30, and dyadic sample sizes are shown in Supplementary Table 51. Our species-specific analyses were restricted to the abovementioned six species categories (or phenotypes) to assure robust statistical assessments. However, other species that were sporadically part of groups were the lionfish *Pterois miles*, the abudjubbe wrasse *Cheilinus abudjubbe*, Klunzinger's wrasse *Thalassoma rueppellii*, the bandcheek wrasse *Oxycheilinus digrammus* and the sand lizardfish *Synodus dermatogenys*. Trajectories of these individuals were still tracked and taken into account when calculating group parameters (for example, when computing group centroids).

## Pull-and-anchor analysis

The main analytical methodology used to quantify leadership was a modified version of the pull-and-anchor analysis[3]. In this paper, GPS tracks were 2D as animals moved almost exclusively in a 2D horizontal plane (that is, baboons moving over $xy$ coordinates), so the code was adapted to 3D to include the $z$ coordinate, that is, the vertical movements of the octopus and fishes in the water column. Compared with other metrics (for example, directional or speed correlations), pull-and-anchor analyses are particularly suited for groups that show erratic movement patterns and frequently change between tight to sparse formations, as well as for analysing movement sequences over short or long timescales[11] (Supplementary Fig. 1).

The pull-and-anchor analysis focuses on assessing variation in dyadic distance between two individuals ($i$ and $j$, extracting successful and unsuccessful initiation events (resulting in pulls and anchors, respectively). In essence, we look for minimum and maximum values of dyadic distance, until a minimum ($t_1$)–maximum ($t_2$)–minimum ($t_3$) sequence is formed. For each of these interactions, there is one initiator and one potential follower: between $t_1$ and $t_2$, the individual increasing the distance relating to the other (let it be $i$ in this example) is the initiator and the other individual (let it be $j$) is the candidate follower. After reaching the maximum dyadic distance ($t_2$), the individual that then shortens the dyadic distance dictates whether this sequence was a successful initiation event (that is, pull) or an unsuccessful initiation event (that is, anchor). Following the example above, if the potential initiator $i$ is the one shortening the dyadic distance to $j$, then this sequence is classified as an anchor event, where $j$ is anchoring $i$. If, on the other hand, $j$ is the one shortening the dyadic distance, that means that $j$ followed $i$ and it is classified as a pull event, that is, $i$ is pulling $j$.

Following the original methodology, several steps were taken to ensure that small-scale variations were not included as pull–anchor events by filtering candidate sequences using disparity, strength and noise thresholds. Before candidate sequences were identified per se, to prevent jittering interference and small body part movements being considered as potential events, a noise threshold of 0.1 m was set as the minimum dyadic distance change between $i$ and $j$. Note that this value is one to two orders of magnitude above the error calculated for the reconstruction and represents approximately 0.5–1 body size of the individuals generally present in the hunts. Disparity was calculated via the difference of covered distance by $i$ and $j$ across each time segment in relation to each other. Complementarily, strength was calculated as the relative change in dyadic distance between each time segment. Both these parameters range between 0 and 1, where, in the case of disparity, values near 0 depict an interaction in which both $i$ and $j$ moved similar distances during each time segment (thus making classification of the event ambiguous), and values near 1 indicate that either $i$ or $j$ performed the majority of movement in each time segment. In the case of strength, values near 0 indicate that the relative change in dyadic distance in each time segment was negligible (that is, $i$ and $j$ were always close together), whereas values near 1 indicate that the distance markedly changed among $t_1$, $t_2$ and $t_3$ (that is, individuals were close, then far apart, then close again). We defined both thresholds at a minimum of 0.1. Using this methodology, we extracted 516 pulls and 664 anchors from the dataset, in a total of 1,180 events.

Pull–anchor metrics were normalized for the number of individuals present in each subgroup and, in species-specific analyses, the number of individuals from each species present. Thus, for each individual $i$, we quantified:

(1) Normalized pulling frequency—number of pulls (that is, successful initiations) per minute ($p_i$)
(2) Normalized anchoring frequency—number of events anchoring an initiator per minute ($a_i$)
(3) Normalized initiation frequency—number of initiations per minute ($p_i + a_i$)
(4) Normalized follower frequency—number of events following when solicited by an initiator per minute ($f_i$)
(5) Normalized anchored frequency—number of anchors (that is, unsuccessful initiations) per minute ($an_i$)

Moreover, we also quantified ratios between leader–follower categories in pull–anchor events by measuring:

(1) Initiation/pulling ratio (pulling efficiency)—number of successful pulling events divided by the total number of initiations
(2) Anchoring/anchoring opportunities ratio (anchoring efficiency)—number of events anchoring an initiator divided by the total number of times solicited
(3) Pulling/following ratio—number of events pulling divided by the number of events following
(4) Initiations/following ratio—number of initiations divided by the number of events following others

See also Supplementary Table 52 for a focus on the metrics that translate to direct influence in others, either by stimulating or inhibiting their movement.

As referred in ref. 11, while observational field data do not explicitly capture causality, it is statistically improbable that all individuals in a group would make the same causally independent decision (despite considerable individual differences), at the same time, across hundreds of decision events, as would be required to explain group cohesion for the duration of hunting scenes. While some level of independent decision-making forms some part of the dynamics we observe, we have taken several steps beyond data thresholding to ensure that the statistical patterns found are consistent, allowing us to state with some confidence that these independent decision-making events become inconsequential to our conclusions.

First, to ensure that individuals were not simply moving randomly in space, we retrieved the angles between the relative movement vectors of the initiator (during $t_1$–$t_2$) and the potential follower (during $t_2$–$t_3$) when pulls and anchors occurred (Supplementary Fig. 19 and see also 1.general script[30]). First, we performed a Rayleigh's test and verified that the frequency distribution is non-uniform, that is, there is preference for movement in certain angle ranges ($P < 0.0001$). From plotting a histogram of the most frequent pull and anchor angles in our dataset (Supplementary Fig. 19), a clear difference in probability emerged, statistically different from what one would expect from a 360° random movement (that is, movement equally likely to happen at any given angle). In our system, anchors are actually 2.3667 times more likely than pulls, if an individual would move in a random angle relatively to another. Moreover, if we calculate the probability of pulling or anchoring for each given angle (rounded as integers between 0 and 359), we can then obtain the average of all probabilities across integers, to get the mean probability of a pull or an anchor occurring at chance level. In our system, the mean probability of pulls is ~31.31% (median probability is even lower: 20%) and that of anchors is ~68.69% (median probability: 80%) (see end of 1.general.ipnyb). Noteworthily as well, the narrow angle range at which pulls occur (relative to any other angle) shows a bimodal high-frequency distribution around 30° and −30°, with a small number of pulls occurring

near 0° (Supplementary Fig. 19). Thus, in the large majority of cases, pulling does not occur owing to movement inertia—that is, individual $j$ apparently follows $i$, but such happens because $j$ was moving in the same direction; individuals deviate from their course and actively follow others.

Moreover, we also ensured that the patterns that emerged from the data were consistent over different levels of organization or potential clustering. First, to prevent our results from being driven by simple preferential association with conspecifics, we analysed pulling frequencies without same-species interactions. Second, we analysed pulling frequencies only when an individual was the first initiator of any given follower. As such, we removed any potential pulls stemming from simply associating with the first initiator strongly. Lastly, we calculated the dyadic pull–anchor interactions between individual $i$ and the centroid of the rest of the group, calculated as the mean position of all individuals except $i$. All results were found to be consistent across the different conditions.

### Individual kinematics and group-level metrics

For each individual trajectory at a given time point, we calculated speed as the difference in the Euclidean distance given by the $xyz$ positions in sequential frames normalized per second and tortuosity as the ratio between the average direction change of the trajectory over 12 frames or 4 s (6 frames or 2 s before and 6 frames or 2 s after) and moving in a straight direction. At the group level, we estimated the centroid of the group as the mean of all individuals present in the subgroup and calculated the Euclidean distances from a given individual and said centroid. Moreover, we also estimated the absolute angle between the individual trajectory and the centroid trajectory between frames as the angle to the centroid of the individual. To explore whether differences in individual angles could impact pull-and-anchor outcomes (successful or unsuccessful), we also calculated the average absolute angle between the initiator's trajectory and the potential follower's trajectory during initiations (between $t_1$ and $t_2$).

### Hierarchical social networks

Social network analysis has been widely used in the fields of sociality and group behaviour over the past decade[45]. This type of analyses comprises a number of different tools that allow characterization of specific individual roles within groups as well as the structure and characteristics of their social interactions with other members of the group[46]. Following refs. 47,48, we adapted this approach by including directed social networks (that is, the influence of individual $i$ over individual $j$, instead of 'dyadic influence' by itself) and by constructing a species rank order that depicts the hierarchical influence in groups.

To find species hierarchical structures within groups across different pull–anchor event parameters, we created adjacency matrixes in which pairwise comparisons between species $l$ and $m$ were performed on both sides of the initiator–potential follower axis, normalized by the number of co-occurrences in subgroups. From these matrixes, we used 2D directed social networks to find underlying network hierarchies for each parameter. First, edge direction (that is, arrow) was determined by the highest positive number, signifying, for example, that in a given interaction, species $l$ was more likely to pull $m$ than the opposite. Next, edge wideness was calculated as the difference between the scores of species $l$ and $m$ in pulling each other. In other words, if individual $i$ had a much higher pulling score than $j$, the edge will be wider and directed to $i$. To clarify absolute hierarchies in each axis, node positions represent a given species' mean ± s.e. parameter score per subgroup, for example, the average of all species-specific pulling scores for the species $m$ as the initiator. Lastly, node colours were calculated as the mean value between $xy$ node positions and portrayed using an $xy$ mixed colour gradient, in which brighter and darker tones, respectively, indicate lower or higher values. All hierarchical social networks were built with package 'NetworkX' in Python.

## Statistics

All statistical analysis were performed in R[49]. We used GLMM available in the package 'glmmTMB' to account for intragroup and intraindividual variability, and used an autoregressive component to deal with temporal correlations. Thus, as a first step, we implemented autoregressive model structures based on the subgroup number within each group, thus not only maintaining identity but also weighing temporal autocorrelation between subgroups within the same group (that is, subgroup + 0|group/individual). When this approach failed, that is, the models failed to converge or were outperformed by simpler GLMMs, we used a nested structure of random effects with individual identity nested within groups, that is, maintained across different subgroups (that is, 1|group/individual/subgroup). For web-over temporal characteristics, we also included field site as the first term of the nested random effects, given additional footage used from Egypt when the octopus was hunting alone.

Statistical models measuring different parameters were implemented as follows. Modelling of continuous variables (for example, speed, distance to centroid) was attempted first using a Gaussian distribution. In case of violation of assumptions, gamma distribution families were then used, first with log link and subsequently with inverse link if necessary. On the other hand, modelling of count data (for example, pulling frequency or anchoring frequency, with the number of individuals present in the subgroup used as an offset) was first attempted using Poisson distributions with identity or log links, depending on data distribution. In case of violation of assumptions or overdispersion, negative binomial distributions were then used. Zero inflation was used for both continuous and count variables in cases of excess of zeros and if necessary to enable model convergence. To measure ratios that constitute binary variables (for example, pull or anchor, within the total number of leadership attempts made), we used binomial distributions. In the case of ratios created from variables that are not binary (for example, the ratio between pulling frequency and anchoring frequency), we used the same approach as modelling count data, but included the second variable (in this example, anchoring frequency) as an offset of the first (in this case, pulling frequency). All models were validated by checking homogeneity of variances, residuals normality and normality of random effects, as well as overdispersion and collinearity when required (for example, with count data and when multiple factors were tested), using the 'check_model' function within the package's 'performance' and 'see'. Pairwise comparisons between levels within significant factors were analysed via Tukey HSD with Tukey multiplicity adjustments considering the number of comparisons made, using the package 'emmeans'.

Finally, to calculate differences in the proportion of number of occurrences and number of punches received (for example, between different species or differently characterized groups), we used first $\chi^2$ tests with continuity corrections measuring the deviation of sampled distributions from random distributions (for example, where punches performed equally to all species) and afterwards performed two-sample equality of proportion tests with continuity correction to test pairwise differences (for example, between different species). Pearson correlations were also calculated to highlight meaningful differences. Results were plotted using 'ggplot2' and visually enhanced using Adobe Illustrator.

## Reporting summary

Further information on research design is available in the Nature Portfolio Reporting Summary linked to this article.

## Data availability

All data used in the analysis are available via figshare at https://doi.org/10.6084/m9.figshare.26214830 (ref. 30).

## Code availability

All code used for analysis is available via figshare at https://doi.org/10.6084/m9.figshare.26214830 (ref. 30).

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

## Acknowledgements

We thank the fieldwork assistants, discussions with colleagues in the Department of Collective Behaviour, and all staff of the Inter-University Institute for Marine Sciences (Israel), Open Ocean Sciences Centre (Egypt) and Lizard Island Research Station (Australia) for invaluable assistance during fieldwork seasons. E.S. acknowledges funding from Fundação para a Ciência e Tecnologia (FCT) with a PhD grant (SFRH/BD/131771/2017) and the MARE strategic project (UID/MAR/04292/2019). Fieldwork was also funded by National Geographic Society (EC-427R-18), PADI Foundation, Malacological Society of London and Animal Behavior Society. V.H.S. acknowledges support from the CRG-CASCB joint grant, funded by the EAS Department, MPI-AB and CASCB at the University of Konstanz. A.S. acknowledges the International Master of Science in Marine Biological Resources (IMBRSea, Ghent University). M.N. acknowledges support from the Hungarian Academy of Sciences, grant 95152 (to the MTA-ELTE 'Lendület' Collective Behaviour Research Group), and the National Research Development and Innovation Office under grant no. K128780. A.S.-P. acknowledges support from the Gips-Schüle Stiftung. I.D.C. acknowledges support from the Office of Naval Research (grant number N00014-64019-1-2556), the European Union's Horizon 2020 research and innovation programme under the Marie Skłodowska-Curie grant agreement (number 860949), and the Deutsche Forschungsgemeinschaft Gottfried Wilhelm Leibniz Prize 2022 584/22 (I.D.C.) All authors acknowledge support from the Deutsche Forschungsgemeinschaft (DFG, German Research Foundation) under Germany's Excellence Strategy-EXC 2117–422037984 (to I.D.C.) and the Max Planck Society.

## Author contributions

E.S., S.G., R.R. and I.D.C. conceptualized the study. Investigation was performed by E.S., V.H.S., A.S. and S.G., and the methodology developed by F.A.F., P.N., M.N., A.S.-P, V.H.S., E.S. and I.D.C. Data visualization was performed by E.S., M.N., V.H.S., I.D.C. and S.G. The project was supervised by S.G., I.D.C. and R.R., and administered by S.G., R.R., I.D.C. and E.S. The original draft was written by E.S. and I.D.C., and all authors reviewed and edited the manuscript.

## Funding

## Competing interests

The authors declare no competing interests.

## Additional information

**Correspondence and requests for materials** should be addressed to Eduardo Sampaio.

[1]MARE—Marine and Environmental Sciences Centre, Laboratório Marítimo da Guia, Faculdade de Ciências, Universidade de Lisboa, Lisbon, Portugal. [2]Department of Collective Behaviour, Max Planck Institute of Animal Behavior, Konstanz, Germany. [3]Centre for the Advanced Study of Collective Behaviour, University of Konstanz, Konstanz, Germany. [4]Department of Biology, University of Konstanz, Konstanz, Germany. [5]Department for the Ecology of Animal Societies, Max Planck Institute of Animal Behavior, Konstanz, Germany. [6]Science of Intelligence (SCIoI), Technische University, Berlin, Germany. [7]Department of Biology, University of Massachusetts Boston, Boston, MA, USA. [8]MTA-ELTE 'Lendület' Collective Behaviour Research Group, Hungarian Academy of Sciences, Budapest, Hungary. [9]Department of Biological Physics, Eötvös Loránd University, Budapest, Hungary. [10]Departamento de Biologia Animal, Faculdade de Ciências, Universidade de Lisboa, Cascais, Portugal. [11]These authors contributed equally: Iain D. Couzin, Simon Gingins. ✉e-mail: esampaio@ab.mpg.de

# Reporting Summary

## Statistics

For all statistical analyses, confirm that the following items are present in the figure legend, table legend, main text, or Methods section.

| n/a | Confirmed | |
|---|---|---|
| ☐ | ☒ | The exact sample size (*n*) for each experimental group/condition, given as a discrete number and unit of measurement |
| ☐ | ☒ | A statement on whether measurements were taken from distinct samples or whether the same sample was measured repeatedly |
| ☐ | ☒ | The statistical test(s) used AND whether they are one- or two-sided *Only common tests should be described solely by name; describe more complex techniques in the Methods section.* |
| ☐ | ☒ | A description of all covariates tested |
| ☐ | ☒ | A description of any assumptions or corrections, such as tests of normality and adjustment for multiple comparisons |
| ☐ | ☒ | A full description of the statistical parameters including central tendency (e.g. means) or other basic estimates (e.g. regression coefficient) AND variation (e.g. standard deviation) or associated estimates of uncertainty (e.g. confidence intervals) |
| ☐ | ☒ | For null hypothesis testing, the test statistic (e.g. *F*, *t*, *r*) with confidence intervals, effect sizes, degrees of freedom and *P* value noted *Give P values as exact values whenever suitable.* |
| ☒ | ☐ | For Bayesian analysis, information on the choice of priors and Markov chain Monte Carlo settings |
| ☐ | ☒ | For hierarchical and complex designs, identification of the appropriate level for tests and full reporting of outcomes |
| ☐ | ☒ | Estimates of effect sizes (e.g. Cohen's *d*, Pearson's *r*), indicating how they were calculated |

*Our web collection on statistics for biologists contains articles on many of the points above.*

## Software and code

Policy information about availability of computer code

| Data collection | To transform 2D videos into 3D reconstructions, we used CVAT 2.3, colmap 3.9, and multiviewtracks 1.0 (https://github.com/pnuehrenberg/multiviewtracks) |
|---|---|
| Data analysis | We used Python 3.8, SciPy and R 4.3.1 to analyze the data. Code used is available at https://doi.org/10.6084/m9.figshare.26214830 |

For manuscripts utilizing custom algorithms or software that are central to the research but not yet described in published literature, software must be made available to editors and reviewers. We strongly encourage code deposition in a community repository (e.g. GitHub). See the Nature Portfolio guidelines for submitting code & software for further information.

## Data

Policy information about availability of data

All manuscripts must include a data availability statement. This statement should provide the following information, where applicable:
- Accession codes, unique identifiers, or web links for publicly available datasets
- A description of any restrictions on data availability
- For clinical datasets or third party data, please ensure that the statement adheres to our policy

All data are available in Figshare, https://doi.org/10.6084/m9.figshare.26214830

# Research involving human participants, their data, or biological material

Policy information about studies with human participants or human data. See also policy information about sex, gender (identity/presentation), and sexual orientation and race, ethnicity and racism.

| | |
|---|---|
| Reporting on sex and gender | na |
| Reporting on race, ethnicity, or other socially relevant groupings | na |
| Population characteristics | na |
| Recruitment | na |
| Ethics oversight | na |

Note that full information on the approval of the study protocol must also be provided in the manuscript.

# Field-specific reporting

Please select the one below that is the best fit for your research. If you are not sure, read the appropriate sections before making your selection.

☐ Life sciences      ☐ Behavioural & social sciences      ☒ Ecological, evolutionary & environmental sciences

For a reference copy of the document with all sections, see nature.com/documents/nr-reporting-summary-flat.pdf

# Ecological, evolutionary & environmental sciences study design

All studies must disclose on these points even when the disclosure is negative.

| | |
|---|---|
| Study description | We recorded collective hunting events of octopuses and multiple fish species over ~120h of diving. We found and 3d reconstructed 13 unique groups that we split into blocks of 100 seconds (the length of the shortest hunting event), thus creating 107 subgroups nested within the first 13. We tested species and individual level differences in kinematics, as well as in dyadic dynamics of pull-anchor interactions (All details for species sample sizes and subgroup compositions are given in DataS1). We further looked at differences in kinematic and pull-anchor properties between groups depending on the presence or absence of the most extreme phenotypes (i.e. species). We then looked at the distribution of partner control mechanisms depending on species and evaluated individual-level behavior from the octopus. we investigated how the frequency and occurrence of partner control mechanisms by the octopus was affected by species presence and group kinematic variables such as displacement. Lastly, we looked at how web-over temporal characteristics changed in accordance to shifts in group composition, using the absence/presence of extreme phenotypes. We complemented these observations by performing a field experiment gauging how web-over temporal characteristics changed depending on the presence of food. |
| Research sample | Octopuses (Octopus cyanea ) and multiple fish species, including : long barbel goatfish Parupeneus macronemus yellow and blue phase gold-saddle goatfish Parupeneus cyclostomus, lyretail grouper Variola louti, and blacktip grouper E. marginatus. |
| Sampling strategy | Using SCUBA, two divers surveyed coral reef areas for collective hunting activity. When found, given that these hunts are not stationary, we adopted a search-and-follow procedure while maintaining a distance of >5 m to minimize disturbing natural interactions. We used two full-frame Sony Alpha 7SII with Zeiss 2/25mm wide lenses mounted on an aluminum structure, as a stereocamera setup (hereafter 'Stereocamera Rig', Fig 1B). A third full-frame Sony Alpha 7SII with Sony f/4 24-70mm lens served as a focal camera for the octopus (hereafter 'Zoom Camera'). Sample size was inherently determined by the biological conditions, and species for which were present n < 5 were excluded from species-level analyses. The 120h of diving and changes in group composition provided a robust  dataset to evaluate the different facets of collective hunting and individual behavior studied. |
| Data collection | Videos from the Stereocamera Rig were processed in our coding pipeline to create 3d reconstructions of individual tracks. Together with an additional ~30 minutes dataset in Egypt (El Quseir 26.1014° N, 34.2803° E), video from the Zoom Camera was used to quantify temporal characteristics of web-overs in solitary octopuses and punching variables. |
| Timing and spatial scale | Fieldwork spanned one month between 01-10-2018 and 01-11-2018 (29.5577° N, 34.9519° E, Eilat, Israel) in a total of ~120h of diving (~60h each diver). Dives were performed 2/3 dives/day, at relatively shallow depths (5-15m) allowing for ~2-3h underwater/day, complying with local scientific diving regulations. A field experiment was performed in Australia between 15-06-2023 and 15-07-2023. |
| Data exclusions | Data were not excluded from the analyses |
| Reproducibility | Analyses were rerun over several versions of R, producing similar results consistently. |
| Randomization | The original hunting groups filmed were divided into 100 second blocks of subgroups. We accounted for this dependence using auto-cirrelative structures to our statistical models whenever models were not outperformed for random effects modelling. |

| | |
|---|---|
| Blinding | Blinding was not possible due to the presence of other animals not being controlled, or possible to be occluded. |

Did the study involve field work? ☐ Yes ☐ No

## Field work, collection and transport

| | |
|---|---|
| Field conditions | Coral reefs, water temperature between 24 and 29 degrees, visibility between 5 and 20 meters. |
| Location | Israel, Egypt, and Australia |
| Access & import/export | na |
| Disturbance | none |

# Reporting for specific materials, systems and methods

We require information from authors about some types of materials, experimental systems and methods used in many studies. Here, indicate whether each material, system or method listed is relevant to your study. If you are not sure if a list item applies to your research, read the appropriate section before selecting a response.

### Materials & experimental systems

| n/a | Involved in the study |
|---|---|
| ☒ | Antibodies |
| ☒ | Eukaryotic cell lines |
| ☒ | Palaeontology and archaeology |
| ☐ | ☒ Animals and other organisms |
| ☒ | Clinical data |
| ☒ | Dual use research of concern |
| ☒ | Plants |

### Methods

| n/a | Involved in the study |
|---|---|
| ☒ | ChIP-seq |
| ☒ | Flow cytometry |
| ☒ | MRI-based neuroimaging |

## Animals and other research organisms

Policy information about studies involving animals; ARRIVE guidelines recommended for reporting animal research, and Sex and Gender in Research

| | |
|---|---|
| Laboratory animals | No laboratory animals were used in this study |
| Wild animals | We observed collective hunts of octopuses (Octopus cyanea) and multiple fish species, including , long barbel goatfish Parupeneus macronemus yellow and blue phase gold-saddle goatfish Parupeneus cyclostomus, lyretail grouper Variola louti, and blacktip grouper E. marginatus. We estimate the age of the subjects between 6 months and 2 years. No direct experimental animal handling was performed. Structures with food and without food were presented to octopuses, following ethics permit from the authorities specified below. |
| Reporting on sex | Octopuses have little sexual dymorphism, so no information regarding sex was collected. |
| Field-collected samples | No field collected samples were used in this study. |
| Ethics oversight | Procedures were approved by the Max Planck Institute of Animal Behaviour, the Department of Agriculture and Fisheries Ethics Committee from Queensland, Australia, and the Great Barrier Reef Marine Parks Authority under permit G23/47925.1. |

Note that full information on the approval of the study protocol must also be provided in the manuscript.

## Plants

Seed stocks

*Report on the source of all seed stocks or other plant material used. If applicable, state the seed stock centre and catalogue number. If plant specimens were collected from the field, describe the collection location, date and sampling procedures.*

Novel plant genotypes

*Describe the methods by which all novel plant genotypes were produced. This includes those generated by transgenic approaches, gene editing, chemical/radiation-based mutagenesis and hybridization. For transgenic lines, describe the transformation method, the number of independent lines analyzed and the generation upon which experiments were performed. For gene-edited lines, describe the editor used, the endogenous sequence targeted for editing, the targeting guide RNA sequence (if applicable) and how the editor was applied.*

Authentication

*Describe any authentication procedures for each seed stock used or novel genotype generated. Describe any experiments used to assess the effect of a mutation and, where applicable, how potential secondary effects (e.g. second site T-DNA insertions, mosiacism, off-target gene editing) were examined.*

