## [Peer Review File · Nature Ecology & Evolution]

Peer Review Information

Journal: Nature Ecology & Evolution

Manuscript Title: Multidimensional social influence drives leadership and composition-dependent success in octopus-fish hunting groups

Corresponding author name(s): Eduardo Sampaio

Editorial Notes:

Reviewer Comments & Decisions:

Decision Letter, initial version:

29th February 2024

Dear Dr Sampaio,

Your Article, "Multidimensional influence drives leadership and composition-dependent success in octopus-fish hunting groups" has now been seen by three reviewers. You will see from their comments copied below that while they find your work of considerable potential interest, they have raised quite substantial concerns that must be addressed. In light of these comments, we cannot accept the manuscript for publication, but would be very interested in considering a revised version that addresses these serious concerns.

We hope you will find the reviewers' comments useful as you decide how to proceed. If you wish to submit a substantially revised manuscript, please bear in mind that we will be reluctant to approach the reviewers again in the absence of the major revisions requested.

All three reviewers comment that the writing is often unclear or opaque, and that this is hindering them from understanding the methods. Additionally, two reviewers comment on some technical issues with the analyses which need further validation and/or additional analyses to ensure the conclusions are supported.

If you choose to revise your manuscript taking into account all reviewer and editor comments, please highlight all changes in the manuscript text file.

* Include a "Response to reviewers" document detailing, point-by-point, how you addressed each referee comment. If no action was taken to address a point, you must provide a compelling argument. This response will be sent back to the referees along with the revised manuscript.

* If you have not done so already we suggest that you begin to revise your manuscript so that it

2conforms to our Article format instructions at <http://www.nature.com/natecolevol/info/final-submission>. Refer also to any guidelines provided in this letter.

[REDACTED]

If you wish to submit a suitably revised manuscript we would hope to receive it within 6 months. If you cannot send it within this time, please let us know. We will be happy to consider your revision so long as nothing similar has been accepted for publication at Nature Ecology & Evolution or published elsewhere.

Nature Ecology & Evolution is committed to improving transparency in authorship. As part of our efforts in this direction, we are now requesting that all authors identified as 'corresponding author' on published papers create and link their Open Researcher and Contributor Identifier (ORCID) with their account on the Manuscript Tracking System (MTS), prior to acceptance. This applies to primary research papers only. ORCID helps the scientific community achieve unambiguous attribution of all scholarly contributions. You can create and link your ORCID from the home page of the MTS by clicking on 'Modify my Springer Nature account'. For more information please visit please visit www.springernature.com/orcid.

Thank you for the opportunity to review your work.

[REDACTED]

Reviewer expertise:

Reviewer #1: collective behaviour and motion

Reviewer #2: sociality, marine biology

Reviewer #3: social network analyses, evolution of collective behaviour

Reviewers' comments:

Reviewer #1 (Remarks to the Author):

In their manuscript, Sampaio et al. describe a study that investigates social hierarchy in mixed-species foraging groups consisting of different fish species and an octopus. They found leadership in these groups to be shared and specific to different stages of the foraging behavior. While the octopus determines when a group is moving towards another foraging site, one species of fish determines the direction of this movement. Up to date, research on leadership in collective behavior is mainly focused on groups consisting of only a single species, and the social hierarchy in mixed-species groups is still poorly understood. Therefore, the present manuscript will contribute heavily to our understanding of the social structure in animal collectives. In general, this is a cool study, which fits very well the scope of Nature Ecology and Evolution. I have some concerns regarding the data analysis and interpretation of the results, but hope the authors will be able to address these concerns. I really would like to see this manuscript published.

1) While reading the manuscript, which by the way is very well written, I struggled most with the method used for analyzing pull- and anchor events. This analysis is the main basis for later conclusions on each specie's role inside the group. Instead of analyzing an individual's influence on movement behavior of the entire group, only interactions between two single individuals within the group were analyzed. This analysis, however, introduces ambiguity and bias to the data, which may affect the conclusions that can be drawn from the results.

a) When analyzing movement initiation – following interactions in dyads of animals, which are actually a fraction of a group, it is often unclear whom the follower is actually following. Imagine individual 1 is moving away from the group, i.e. is attempting to initiate a group movement, and individual 2 is following individual 1. When now the interaction between individual 2 and a third individual is analyzed, it is unclear whether a following movement of individual 3 was caused by individual 2 or by individual 1, i.e. whether individual 3 is indeed following individual 2 or is actually following individual 1. Assuming that in such a scenario, the interaction between individual 2 and individual 3 is a pull event, would be incorrect, because the movement in individual 3 might have been initiated by individual 1.

b) The analysis was not only performed on dyads of individuals belonging to different species, but also on dyads of individuals of the same species. Pull and anchor events in same-species dyads can be driven by specie-specific traits that don't play a role in interactions of mixed-species dyads. In the present study, blue goatfish seem to pull other blue goatfish very frequently (Fig. 2C). In Steinegger et al. 2018 Proc Biol Sci, it is demonstrated that gold-saddle goatfish form collaborative hunting groups and follow each other when one individual leaves the group. Gold-saddle goatfish thus seem to have a species-specific drive to follow each other, which biases the analysis on pull frequency. It is important that the method used to analyze movement interactions in the group takes account of differences in species-specific traits and intraspecies interactions between members of the group. Especially, when for some species in the group, such as the octopus, always only a single individual was present in the group, and within-species interactions, such as in blue goatfish, were impossible.

To reduce ambiguity and bias, I suggest to apply the following changes to the analysis of pull-anchor

3events:

Instead of analyzing any variation in distance between two individuals of the group, only variation in distance between an individual that attempts to initiate a group movement and the centroid of the other group members should be analyzed. An attempt to initiate a group movement could for example be defined as a significant deviation of an animal's position from the edge of the area a group is occupying for a given time period. When the group of remaining animals is moving towards the individual that attempted to initiate a group movement and the distance between the group's centroid and this animal is decreasing after an initial increase, the initiation attempt was successful and can be counted as a pull event. If the group of remaining animals stays put and the animal that attempted to initiate a group movement moves back towards the group and hence causes the distance between itself and the centroid of the group of remaining animals to decrease after an initial increase, the initiation attempt was unsuccessful and can be counted as an anchor event. The individual of the group of remaining animals that changed its position least between time points t_2 and t_3 can be considered the anchor in this specific anchoring event. This type of analysis will ensure that indeed the influence of one individual on the entire group is measured, will prevent false conclusions on who is following whom and will reduce the impact same-species interactions will have on the results.

2) To estimate the strength of influence a certain species has on the multi-specific group, the normalized values for pulling and anchoring frequency were summed up for each species. This is indeed the classical way to define leadership in same-species collectives, but is not applicable to multi-specific groups where species-specific traits come into play. Mobility, for instance, is a species-specific trait. A species that moves a lot in general will also move away from the group more often than a species that is sedentary in general. In this scenario, high general mobility should not be confused with a strong drive to elicit movement in others. To reduce the influence of species-specific traits, the efficiency (leader quality) rather than the frequency of actions with which a species is influencing the movements of others should be used to estimate a species' general influence and its role in a multi-species group. Blue goatfish, and goatfish in general, attempted to initiate movement in others most frequently and also pulled other individuals most frequently. The latter, however, is simply due to the goatfish' high rate of initiation attempts. Although, blue goatfish seem to influence others most frequently, their initiation attempts were successful in on average only 43% of attempts. This is below chance level. Attempting to influence others very often and being lucky to be successful in less than half of the attempts, likely indicates that the goatfish was not considered a reliable leader by the other group members. The octopus and the blacktip, however, were able to influence the movement of others in on average 70% and 60% of attempts, respectively, and were the only species that were able to consistently initiate or inhibit movement in the majority of the other species. I agree that leadership is shared in octopus-fish hunting groups, but rather between the octopus and the blacktip instead of the octopus and the blue goatfish.

3) This manuscript has the potential to be of interest for a broad range of readers. To increase its appeal, the manuscript may benefit from adding a few basic details.

a) The abstract, for instance, is missing a clear definition of the scientific question the study aims to answer. In the first three sentences of the abstract, the motivation for the study is explained, followed directly by a description of the methods and the results. Telling the reader what exactly was this study's aim, might help them to judge the importance of the findings.

4- b) The last sentence of the abstract lists various aspects of collective behavior, for which the authors think their study is increasing our understanding. A reader who is not very familiar with the listed concepts, might not know how exactly the study contributes to the understanding of any of them. Instead of just mentioning that the study furthers understanding of A, B, C, etc., I suggest to pick one or two of these aspects and explain how the study is adding knowledge to these areas.
- c) In the introduction, it might be helpful to mention explicitly, that the octopus is the main driver of the hunting group's formation and is therefore considered the nuclear species (see Mangini et al. 2023 Philos Trans R Soc Lond B Biol Sci). Or do the five fish species also form hunting groups without an octopus?

Line specific comments:

1) Lines 149 – 153: Can you include a hypothesis on why blue goatfish are the main drivers of movement initiation?

2) Lines 209 – 211: Might it also be possible that the main function of the blacktip is to maintain group cohesion? Was web-over frequency increased in groups without blacktips? If so, this would indicate a negative influence of increased group cohesion on environmental exploration and consequently hunting success in groups with blacktips present. Instead, an increased group cohesion could also have positive influences on the group's success, e.g. by facilitating predation avoidance.

3) Line 291 – 292: The number of punching events is significantly lower when blue goatfish are present than when blue goatfish are absent. You conclude from this fact that the presence of blue goatfish stabilizes the social network. However, the reduction in punching probability when blue goatfish are present, can also be caused by the fact that blacktips were often absent at the same time when blue goatfish are present. In 67% (39 of 58) of the subgroups in which blue goatfish were present, blacktips were absent. This means that in the majority of subgroups with blue goatfish present, the main recipient of punching, the blacktip, was not there, which most likely affected the probability of punching more strongly than the presence of blue goatfish.

4) Lines 333 – 336: The negative slope between web-over duration and web-over frequency is likely caused by the natural interaction between feeding time and feeding frequency. An octopus that spends a long time feeding at one food source, is able to visit a lower number of food sources in a given time window than an octopus that stays at one food source only for a short time period. Might it be that with the help of the blue goatfish the octopus is able to find food sources of higher quality than when foraging alone? Exploiting these high-quality food sources takes longer than exploiting low-quality food sources. Consequently, the octopus can only visit a low number of high-quality food sources within 100 seconds. If the increase in prey-finding opportunities due to an increased radius for environmental exploration when the blacktip is absent is causing the increase in feeding success in the octopus, the frequency of web-overs should increase with an increase in the number of potential food sources.

5) Lines 485-486: The word "of" is missing. Multi specific hunting groups can be composed of several species, ...

56) Line 487: Please write out the genus name for the blacktip grouper. Short names are not in bold, as stated in the brackets.

7) Fig. 2: I like the directed hierarchical social network plots. They convey an immense amount of information. However, I don't understand the meaning of the letters a, b, c. Significance between which parameters do they indicate? I further suggest to remove the coloring of the nodes. The same information that is conveyed by node color is also provided by the relative position of each species' picture (node) in the coordinate system. The different node colors are only distracting. Especially the octopus in Fig. 2B, 2C and 2D is hard to see because the node color is very dark. What is indicated by the asterisk in the box next to the octopus in 2C? And, what is meant with the error bar and asterisk in the middle of 2D? Shouldn't the red arrows in 2C actually be yellow, as they indicate frequency as in 2A and not efficiency as in 2B?

8) Line 509: "Group composition" instead of "Species composition"?

9) Line 518: Remove "the" between "combined" and "factorial".

10) Fig. 4: The photos in A and B are meant to demonstrate dynamic events. Dynamic events are hard to see in a single image. It may help to provide a series of images instead. The labels "A" and "B" are almost not visible. Maybe changing the font color to white will help. Could you please indicate subgroups with and without blacktips in E by colored symbols? I guess that group composition, especially the presence of blacktips, in addition to centroid displacement is affecting the probability of punching.

11) Supp. Material Line 86: please explain what the reason was for dividing the hunting scenes into blocks of 100 seconds. How was that done, simply by cutting each scene after every 100 seconds, or were specific starting points defined for each 100-second block?

12) Supp. Material Line 87: Please indicate of how many different octopus individuals the 107 subgroups consisted. Please also mention the average number of group members and the average group composition.

13) Supp. Material Table at Line 164: In the description of the Interpretation of Efficiency for Anchoring, "(probability of success)" should read "(probability of failure)", shouldn't it?

Reviewer #2 (Remarks to the Author):

NATECOLEVOL-23123085-T

Multidimensional influence drives leadership and composition-dependent success in octopus-fish hunting groups

This is a valuable study. The authors have completed an excellent job at demonstrating the different roles within this unique multispecies hunting group. The authors test whether octopuses and various

6fish species drive leadership by using a pull-and-anchor analysis. Importantly, the authors include jittering interference by using a noise threshold of 0.1m. By adding this small amount of random variation to the data, the authors can reduce random fluctuations that might obscure the true patterns of leadership within the hunting group. The findings of this paper offer insight into how a typically solitary species, the octopus, can demonstrate significant behavioural flexibility within a social setting. This enhances our understanding of the ways in which the behaviour of solitary species can be influenced by social pressures in a heterospecific context.

My concerns are minor and listed below. They primarily pertain to the sentence structure and language used in the manuscript. In several sections, the sentences are excessively lengthy and overly technical. I've provided suggestions aimed at enhancing the manuscript's readability and accessibility. While these concerns are not major, addressing them is necessary before I can endorse this work for publication.

Main text

Line 38: Would the term 'initiating' be more appropriate than 'inducing' here?

Line 39: It's unclear to me whether you mean that 'classical leadership fails to accurately describe ALL complex heterogenous systems or just this one between the day octopus and different fish species. Please clarify in the text.

Line 44-46: I'm getting a little lost here... can you rewrite please; also is the term 'same-species' necessary to include here?

Line 46: Please remove the term 'comprehensively', no need for it.

Line 49: 'constituent components'... seems excessively complex and hard to read. Can't you just say participants instead? I would break down this entire sentence. For e.g. Collective behavior is the result of dynamic interactions between individual parts. It's vital for the coordinated functioning of various systems, whether they are physical, cellular, or social.

Line 52: what do you mean by 'informational differences'?

Line 53-56: Similar comment as above. This sentence uses a lot of technical terms and is a lot to process. I would reframe. For e.g. We see this complex behaviour in actions such as the division of labour among cells and in insect colonies, as well as in how groups make decisions. These behaviours open up various leadership possibilities, from having a single leader to sharing leadership responsibilities among many.

Line 65: consider removing the term 'consequent'

Line 83: consider removing the term 'concurrently'

Line 84: please reframe the last part of the sentence i.e., with suggestions that the octopus might

7either disregard the fish or be exploited by them.

Line 113: would the term 'efficacy' (instead of efficiency) be better here, as I believe you're interested in the ability to produce a desired or intended result?

Line 119-126: Really nice explanation of the 'pull' and 'anchor' process.

Line 156-160: Please divide this into two sentences. Perhaps start a new sentence after 'movement in others'

Line 190-194: Consider breaking these two sentences in two. For e.g., the current findings reveal that groups made up of individuals from different species, which are not related and have evolved differently, can achieve complex functions. These functions are similar to those qualitatively described in groups of related individuals from the same species, who exhibit high levels of social interaction.

Lines 229-232: Odd that you only report octopus pulling success here, what about the aforementioned barbel goatfish?

Lines 240-246: Can you please reword; I get lost here.

Lines 289-291: Such a great finding!

Lines 300-304: long sentence, please trim down. You can remove 'in order' for instance and remove terms like 'considerable'.

Lines 313-318: Clever!

Line 333-334: 'Indicating' used twice in quick succession.

Line 340-343: This hurts to read. Try this: In terms of ecological functionality, combining the identified patterns of movement influence and the changes in foraging strategy and benefits reveals specialized hunting roles within the group. The goatfish explore and identify various prey locations, thereby providing options, while the octopus predominantly makes the decision on which options to pursue and attempts to capture the prey. These joint actions result in higher prey capture success rates.

Lines 347-348: This is really interesting, could you elaborate or provide an example of where we might see this in the animal kingdom (outside of the octopus-fish hunting interaction)?

Lines 349: isn't it closer to 550 mya?

Line 359: I think you can drop the word comprehensively.

Line 496: It might be my screen but the colour looks more like purple rather than blue.

Supplementary File

Line 87: A little confused here, how was there a total of 107 subgroups but within that, you've created 130 blacktips sequences? Were the subgroups divided further?

Line 125: Great and important that you included this.

Reviewer #3 (Remarks to the Author):

The key contribution this paper is claiming is that three components of leadership on movement decisions (where, when, and if, to move) can be split across different individuals within a collectively moving group. This is a really exciting and important suggestion, and the authors have gone to enormous analytical effort to provide supporting evidence for it. Unfortunately they are slightly let down by the writing in the paper: it is needlessly obfuscating in many parts, to the point of being distractingly so in many places, which did make it hard to review and assess. I do hope they make some effort to improve clarity in many places as it is hard enough to understand for native English speakers, let alone non-English speakers. After reading this paper thoroughly, I'm afraid that I do have two main issues that do need clarification/additional analyses to resolve.

Firstly, one of my main questions concerns the 'pull-and-anchor' metric used by the authors, which was made initially tricky to understand because I cannot find any reference to the sample size of dyads used in their analysis (the supplementary data files referred to are not included in the reviewer documents). Specifically, it wasn't clear how the authors chose each dyad for each event when calculating this metric, particularly individual 'j' (the individual that responds to an initiator), and most importantly when this individual was classified as an "anchor". We have to assume that the authors are copying this component of the methodology used by Strandburg-Peshkin & Farine et al (2015), which they reference under the caveat of making several changes, where for each 'event' the authors created dyads between the single 'initiator' and every other individual within the social group. I find this problematic since the language used by the authors throughout the paper (e.g. "initiation attempt", "pulling frequency" etc), and their analyses, makes the implicit assumption that a dyad constitutes a causal event where a social decision to follow or not is being made between two specific individuals. However, in the majority of 'events', this is not actually true since the information about the 'true dyad' (i.e. the specific individual/individuals an animal is choosing to associate with) will be swamped by the data generated from every other possible connection between the initiator and all other group members. There is little intuitive predictions that can be made about how this will affect the author's results, except that it will likely over-inflate the calculated importance of the non 'true' leaders.

9The authors seem to (ambiguously) imply this on lines 164-167, where they suggest the strong result showing blacktips are highly efficient pullers and anchorers might be caused by a “signal-to-noise ratio” issue. Beyond being completely unclear what this actually means (they really need to change the language here) the only reason they seem to suggest this is because this result did not follow their predictions. Indeed, I assume that the explanation the authors are attempting to make is that blacktips are being classified as highly efficient anchorers and pullers purely because they are associating more strongly with the octopuses than with any other fish, and the other fish within the social group are actually only (or almost entirely) paying attention to the octopuses when deciding whether to return after an “initiation attempt”. This issue with the non-specific classification of social relationships, coupled with the correlational nature of the study, do make me question how much of the other main results are also artefacts. I don’t think this was an issue in Strandberg-Peshkin’s study since they tested directly whether following events were being driven by single vs multiple individuals, but I am not really sure how this issue can be resolved here: to really assess the bias created from this metric, some form of simulation study will be needed where the metrics calculated for true leaders can be compared to that of individuals that simply associate very strongly with leaders. If the authors cannot validate this metric in this manner, then they must explicitly state the shortcomings of this metric so a reader is made aware of its issues, and to change the language used throughout the study to something more appropriate for its purely correlational nature.

Secondly, this brings me onto the octopuses, and whether they are indeed following other fish to the extent that the author’s conclusions in the abstract “specific fish species (particularly goatfish) drive environmental exploration, deciding where [...] the group moves” are valid. The major limitation with this claim is that the efficiency by which octopuses do actually get ‘pulled’ by goatfish is so low, as is the goatfish’s ‘general influence’ on collective movement, that I don’t find this nearly convincing enough to justify the claims made in the paper about leadership being multidimensional. When assessed in concert with the ambiguity of what the pulling/anchoring metrics are telling us, it is conceivable that these results could be obtained if goatfish use the same information as octopuses about where to forage (e.g. the quality of foraging sites via olfactory or visual means), and are just more likely to be at these before an octopus makes its own independent decision, due to differences in mobility and tendency to practice their own independent foraging relative to other fish. The key result needed here is whether octopuses use social information in preference to personal information about where to go when there are multiple equally-suitable options to forage. Beyond an experiment in a standardised foraging arena, I wonder if the authors can provide something approximating this analysis by using their 3D scenery reconstructions to a-priori objectively identify potential foraging sites (e.g. large rocks)? If so, it would be very convincing (to me) if they can then directly test the importance of various environment features between multiple potential foraging sites (e.g. rock size, adjacency, deviation from previous path angle etc) versus social information (if and what species of fish are foraging there).

Minor comments

Much of the language used to describe behaviours is inappropriately leading. For example, there is no evidence that the movements by any of these animals are “initiation attempts” (which implies intent and function): it’s much more parsimonious to explain this behaviour as animals moving off to forage

10nearby and then returning if they are followed: there is no need to infer their 'intentions', so please replace throughout.

Line 46: "comprehensively broaden our understanding of sociality, specialization, self-organization, problem-solving, control theory, and leadership, in collectives across biological scales"

I appreciate the author's enthusiasm for their work, but this is so overtly-aggrandising that I suggest they change it to a more appropriate and level-headed statement to avoid putting readers off.

Line 196-199: here the authors are comparing observational differences in network structure between groups of animals as being akin to gene knockout studies. Again, there is no need for this level of grandiosity: please remove. In addition, I am familiar with the social network literature, and the term "to highlight how node removal can lead to network structure rearrangement" suggests that nodes were actually removed (i.e. it is an experimental removal study: I had to check the supplementary material to doublecheck this is false), when in fact there was no such thing. This part of the study needs to be clearly communicated as purely observational.

Line 212: Where is the evidence that blue goatfish "play a key role in both group mobility (considerably impacting movement decisions)"? They do have an important role in pulling frequency, but their overall "general influence" is not significantly different to other species according to Fig 2D.

Line 266-300: I appreciate how interesting this behaviour is, but it does seem to come out of nowhere, and doesn't seem to be contributing to the key messages made by this paper. Would it be more appropriate being written up as a separate manuscript on its own (to avoid being lost in the main points of this manuscript)?

Line 300 onwards: a potential confound of this analysis is if octopuses are also more likely to web-over to avoid their captured prey being stolen, i.e. it is primarily used for capturing prey, but extended periods of webbing are also used to avoid already-immobilised prey that are being consumed from being stolen. I base this entirely on my own anecdotal observations that octopuses can sometimes move around seemingly-normally whilst in the process of consuming a captured prey item (implying that webbing-over is not purely associated with prey handling and consumption). Can you authors please give a compelling reason why this cannot explain the fact that extended webbing over events occur when there are potential kleptoparasites around?

There is no Data S1 file provided for reviewers to assess.

Author Rebuttal to Initial comments

Response Letter

Reviewer #1 (Remarks to the Author):

11R1.1. In their manuscript, Sampaio et al. describe a study that investigates social hierarchy in mixed-species foraging groups consisting of different fish species and an octopus. They found leadership in these groups to be shared and specific to different stages of the foraging behavior. While the octopus determines when a group is moving towards another foraging site, one species of fish determines the direction of this movement. Up to date, research on leadership in collective behavior is mainly focused on groups consisting of only a single species, and the social hierarchy in mixed-species groups is still poorly understood. Therefore, the present manuscript will contribute heavily to our understanding of the social structure in animal collectives. In general, this is a cool study, which fits very well the scope of Nature Ecology and Evolution. I have some concerns regarding the data analysis and interpretation of the results, but hope the authors will be able to address these concerns. I really would like to see this manuscript published.

REPLY: We thank the reviewer for their appraisal of the manuscript, as well as for the thoughtful and constructive comments

R1.2. 1) While reading the manuscript, which by the way is very well written, I struggled most with the method used for analyzing pull and anchor events. This analysis is the main basis for later conclusions on each specie's role inside the group. Instead of analyzing an individual's influence on movement behavior of the entire group, only interactions between two single individuals within the group were analyzed. This analysis, however, introduces ambiguity and bias to the data, which may affect the conclusions that can be drawn from the results.

REPLY: We thank you for appreciating the writing of our manuscript. As suggested, we have now performed additional analyses employing variations of the pull-anchor method, as we detail below.

Importantly, these analyses all support the conclusions of the original manuscript, demonstrating that our findings are robust. While there exist slight differences in statistical significance levels, these appear to be an inevitable result of sample size differences rather than to differences in effect sizes, as can be seen in the respective figures and tables (specific information detailed below).

These additional analyses are now included in the supplementary information, and are directly referred in the main text, at the beginning of pull-anchor results (L146 to L174). As response to specific comments made by the reviewer, we provide further details of each analysis variant below.

R1.3. a) When analyzing movement initiation – following interactions in dyads of animals, which are actually a fraction of a group, it is often unclear whom the follower is actually following. Imagine individual 1 is moving away from the group, i.e. is attempting to initiate a group movement, and individual 2 is following individual 1. When now the interaction between individual 2 and a third individual is analyzed, it is unclear whether a following movement of individual 3 was caused by individual 2 or by individual 1, i.e. whether individual 3 is indeed following individual 2 or is actually following individual 1. Assuming that in

such a scenario, the interaction between individual 2 and individual 3 is a pull event, would be incorrect, because the movement in individual 3 might have been initiated by individual 1.

REPLY: Indeed, in these scenarios, it is hard to disentangle which individual (1, 2, or a combination of both) was the most influential in pulling individual 3. We believe this is a general issue with all (observational) studies of influence and leadership in social groups, and we are not aware of any existing methodology that can fully address this issue. However, as the reviewer did, one can assume that the 1st individual moving will be the most influential of the pullers. To prevent possible artefacts coming from scenarios such as the one outlined by the reviewer (i.e. individual 1 pulls individual 2 and individual 3, but individual 2 also 'gains' a pull event as a byproduct because it moved with individual 1 before), we re-analyzed our dataset considering only the 1st puller of any given potential follower. That is, we looked for events where multiple animals pulled a given follower in sequence and retained only the first puller, thus eliminating any potential 'byproduct' pulls. The results of this analysis, included in Table S11 and Fig. S3, can be seen to be highly consistent with the results obtained from analyzing the entire dataset, seen in Fig. S2A and Table S2, and are now part of the main manuscript (see also Methods L553-561):

Functionally, goatfishes (particularly blue goatfish individuals) emerge as the main drivers of movement initiation, i.e., their frequency of initiations and pulls per minute (Fig. 2A) being far greater than non-goatfish species, including the octopus (Tukey Honest Significant Differences with correction, all comparisons $p < 0.05$, Fig. 2C, Table S2-3). Similar results are found when analyzing only the 1st puller of a given follower (i.e. the 1st individual moving in pulls where multiple individuals subsequently move, Fig. S3, Table S11) (L143-148)

R1.4. b) The analysis was not only performed on dyads of individuals belonging to different species, but also on dyads of individuals of the same species. Pull and anchor events in same-species dyads can be driven by specie-specific traits that don't play a role in interactions of mixed-species dyads. In the present study, blue goatfish seem to pull other blue goatfish very frequently (Fig. 2C). In Steinegger et al. 2018 Proc Biol Sci, it is demonstrated that gold-saddle goatfish form collaborative hunting groups and follow each other when one individual leaves the group. Gold-saddle goatfish thus seem to have a species-specific drive to follow each other, which biases the analysis on pull frequency. It is important that the method used to analyze movement interactions in the group takes account of differences in species-specific traits and intraspecies interactions between members of the group. Especially, when for some species in the group, such as the octopus, always only a single individual was present in the group, and within-species interactions, such as in blue goatfish, were impossible.

REPLY: The reviewer makes a great point. To address this, we re-analyzed our dataset by removing all same species interactions. The results can be seen in Table S13 and Fig. S5. As above, this analysis also produces similar results to those obtained from analyzing the entire dataset, seen in Fig. S2A and Table S2, demonstrating the robustness of the findings. Thus, we conclude that while blue goatfish do pull each other frequently, they still pull other members of the group very frequently as well, and remain as the main pullers in the group. We have also added the information that gold-saddle perform collaborative hunting to the main manuscript (see also Methods L553-561):

13Moreover, the same pulling influence outcomes are found even when intraspecific interactions are removed (Fig. S5, Table S13). Goatfish, particularly blue and yellow (same species), are active predators with high mobility that find and corner prey together with other conspecifics³². These fish seem to use similar strategies in interspecific groups, serving as social cues for others. However, while blue goatfish have the highest pulling influence, equally mobile yellow goatfish (presumably a younger phase of the same species) exert less influence on the movement of others. (L150-156)

R1.5. To reduce ambiguity and bias, I suggest to apply the following changes to the analysis of pull-anchor events:

Instead of analyzing any variation in distance between two individuals of the group, only variation in distance between an individual that attempts to initiate a group movement and the centroid of the other group members should be analyzed. An attempt to initiate a group movement could for example be defined as a significant deviation of an animal's position from the edge of the area a group is occupying for a given time period. When the group of remaining animals is moving towards the individual that attempted to initiate a group movement and the distance between the group's centroid and this animal is decreasing after an initial increase, the initiation attempt was successful and can be counted as a pull event. If the group of remaining animals stays put and the animal that attempted to initiate a group movement moves back towards the group and hence causes the distance between itself and the centroid of the group of remaining animals to decrease after an initial increase, the initiation attempt was unsuccessful and can be counted as an anchor event. The individual of the group of remaining animals that changed its position least between time points t_2 and t_3 can be considered the anchor in this specific anchoring event. This type of analysis will ensure that indeed the influence of one individual on the entire group is measured, will prevent false conclusions on who is following whom and will reduce the impact same-species interactions will have on the results.

REPLY: We understand the reasoning of the reviewer, considering dyadic interactions (despite normalizing by number of individuals in the group), instead of the group itself, could be a potential confound. To address this concern, we re-analyzed our entire dataset looking for pull-anchor interactions between a given individual i and the centroid of the rest of the group (calculated as the mean position of all individuals, excluding i). Considering pulling, we see that blue goatfish are the main drivers of movement of the group centroid, pulling it more frequently than other species (Fig S4A and Table S12). Across the entire dataset, multiple animals can remain relatively stationary during these initiation events, which makes it difficult to pinpoint which individual(s) anchor the group when anchors occur. However, we solved this issue by directly looking at interactions where the group centroid moved away and individual j can be identified as anchoring the centroid, due to the group turning back and moving to individual j 's position. Considering all of these situations, we find that the octopus is the individual that most frequently anchors the group's centroid (Fig. S4B and Table S15, between L145-172). Thus, the conclusions of our original version of the manuscript are maintained (goatfish drive movement, the octopus decides if and when to move), demonstrating that there are no differences in interpretation that arise from quantifying movement at a group or dyadic level in the way we did, i.e. by normalizing metrics for the number of

individuals within a group. See references in the main manuscript and methods (valid for the reviewer's previous points as well) below:

Similar results are found when analyzing only the 1st puller of a given follower (i.e. the 1st individual moving in pulls where multiple individuals subsequently move, Fig. S3, Table S11) and pulling frequency of the group's centroid (Fig. S4A, Table S12), confirming that these fish are the main drivers of group movement. (L146-150)

Nonetheless, octopuses were the main functional anchorers in multispecific groups, i.e., frequency of events anchoring other individuals per minute (Tukey HSD, all $p < 0.05$, Table S6, Fig. 2C), representing the main driver of movement inhibition in others (and the group's centroid; Fig. S4B, Table S15). (L172-L175)

Moreover, we also ensured that the patterns emerged from the data were consistent over different levels of organization or potential clusterings. First, to prevent our results from being driven by simple preferential association with conspecifics, we analyzed pulling frequencies without same-species interactions. Second, we analyzed pulling frequencies only when an individual was the first initiator of any given follower. As such, we removed any potential pulls stemming from simply associating with the first initiator strongly. Lastly, we calculated the dyadic pull-anchor interactions between individual i and the centroid of the rest of group, calculated as the mean position of all individuals except i . All results were found to be consistent across the different conditions. (L552-560)

The next comment is a complex paragraph, and as such we will break it down to answer each point raised.

R1.6. 2) To estimate the strength of influence a certain species has on the multi-specific group, the normalized values for pulling and anchoring frequency were summed up for each species. This is indeed the classical way to define leadership in same-species collectives, but is not applicable to multi-specific groups where species-specific traits come into play.

REPLY: Just to reiterate, we have shown that there are no differences in interpretation that arise when having multiple individuals of the same species present, regarding the outcomes of the pull-anchor analyses (Fig. S5, see R1.4).

R1.7. Mobility, for instance, is a species-specific trait. A species that moves a lot in general will also move away from the group more often than a species that is sedentary in general. In this scenario, high general mobility should not be confused with a strong drive to elicit movement in others. To reduce the influence of species-specific traits, the efficiency (leader quality) rather than the frequency of actions with which a species is influencing the movements of others should be used to estimate a species' general influence and its role in a multi-species group. Blue goatfish, and goatfish in general, attempted to initiate movement in others most frequently and also pulled other individuals most frequently. The latter, however, is simply due to the goatfish' high rate of initiation attempts.

REPLY: We understand the reasoning of the reviewer, which is why we estimate both the efficiency of pulls as well as the frequency of pulls. Considering the data in light of each offers complimentary interpretations. In terms of the functional consequences to group motion, however, we must consider what has more impact in the group's motion. For example, while the octopus is very efficient when it moves (i.e., very effective per initiation), it does so very infrequently. Therefore, if it depends on the octopus, the group will move very little (see, for example, comparisons to groups in which there are only blacktips and long-barbel goatfish present, Fig. 3). Only when goatfish are present, particularly blue goatfish, does the group significantly increase its displacement. Therefore, functionally, the blue goatfish are more responsible for driving the movement of others (and, consequently, of the group) than is the octopus.

R1.8. Although, blue goatfish seem to influence others most frequently, their initiation attempts were successful in on average only 43% of attempts. This is below chance level. Attempting to influence others very often and being lucky to be successful in less than half of the attempts, likely indicates that the goatfish was not considered a reliable leader by the other group members.

REPLY: Thank you for pointing out this issue, which has highlighted a lack of clarity in the previous version of our Methods. The issue here is assuming that the binary outcome of an initiation is equally likely (50% pull - 50% anchor). However, that is far from the case as anchors and pulls are not random events. For instance, let us consider the angles at which the relative movement of puller(t1-t2)-follower(t2-t3) dyads produce either a pull or an anchor. First, if we perform a Rayleigh's test, we verify that the frequency distribution is non-uniform, i.e. there is a preference for movement in certain angle ranges ($p < 0.0001$). From plotting a histogram of the most frequent pull/anchor angles in our dataset (Fig. S19), a clear difference in probability emerges, statistically different from what one would assume to be a 360 degrees random movement (i.e. equally likely to happen at any given angle). From here, we also calculated that, in our system, anchors are actually 2.3667 times more likely than pulls. Second, if we then calculate the probability of pulling or anchoring for each given angle (rounded as integers between 0 and 359), we can obtain the mean of all probabilities across integers to get the probability of a pull/ or an anchor occurring by 'chance'. In our system, the mean probability of pulls is around ~31.31% (median probability is even lower, 20%) and of anchors is ~68.69% (median probability: 80%) (see end of 1.general.ipnyb). These are the 'null values' for probabilities in our system, not 50%/50%. Thus, if blue goatfish have an average success of 43% on attempts, that is roughly 1.5 times more than chance level in our dataset. Moreover, another interesting feature of this analyzes is the given frequency distribution of pull angles. The narrow angle range at which pulls occur shows a bimodal frequency distribution around 30 and -30 degrees, and a relatively small number of pulls occurring near 0 degrees. Thus, pulling does not occur due to movement inertia — i.e. individual i apparently follows j but such happens because i was already moving in the same direction —, but mainly due to individuals deviating from their course, actively following others. We will refrain from pasting the text here to prevent being too repetitive, but we have written this analysis and explanation in the Methods section, L533-551.

R1.9. The octopus and the blacktip, however, were able to influence the movement of others in on average 70% and 60% of attempts, respectively, and were the only species that were able to consistently initiate or inhibit movement in the majority of the other species. I agree that leadership is shared in octopus-fish hunting groups, but rather between the octopus and the blacktip instead of the octopus and the blue goatfish.

REPLY: Regarding leadership being measured as frequency or efficiency, see the reply to R1.7 concerning the interpretation of metrics. Functionally, blacktip groupers have little to no impact on the movement of others, as shown in Fig 2C. However, we do agree that it is important to highlight the different efficiencies in driving others, which shows that some species (blacktips and octopus) are especially salient (higher signal-to-noise ratio) for others when on the move (possibly suggesting a higher probability of prey near where they move). For that reason, we opted to include these different metrics in the paper (and hence, 'multidimensional influence' not only referring to pulling and anchoring, but also to frequency and efficiency). They are not mutually exclusive, but complementary visions of what are important leadership measurements, but in the end, we argue that the most important facet of leadership in a group, is the one with most functional consequences.

R1.10. 3) This manuscript has the potential to be of interest for a broad range of readers. To increase its appeal, the manuscript may benefit from adding a few basic details.

a) The abstract, for instance, is missing a clear definition of the scientific question the study aims to answer. In the first three sentences of the abstract, the motivation for the study is explained, followed directly by a description of the methods and the results. Telling the reader what exactly was this study's aim, might help them to judge the importance of the findings

REPLY: Rephrased to make the objectives of the study clearer.

Collective behavior, social interactions, and leadership in animal groups are often driven by individual differences. However most studies focus on same-species groups, where individual variation is relatively low. Multispecies groups however, entail interactions among highly-divergent phenotypes, ranging from simple exploitative actions to complex coordinated networks. Here, we studied hunting groups of otherwise-solitary *Octopus cyanea* and multiple fish species to unravel hidden mechanisms of leadership and associated dynamics in functional nature and complexity, when divergence is maximized. (L25-31)

R1.11. b) The last sentence of the abstract lists various aspects of collective behavior, for which the authors think their study is increasing our understanding. A reader who is not very familiar with the listed concepts, might not know how exactly the study contributes to the understanding of any of them. Instead of just mentioning that the study furthers understanding of A, B, C, etc., I suggest to pick one or two of these aspects and explain how the study is adding knowledge to these areas.

REPLY: We have rephrased to focus on more specific aspects addressed in the study.

Thus 'classical leadership' can be insufficient to describe complex heterogeneous systems, where leadership instead can be driven by both stimulating and inhibiting movement. Furthermore, group composition altered individual investment and collective action, triggering partner control mechanisms (i.e., punching) and benefits for the *de facto* leader, the octopus. This seemingly non-social invertebrate flexibly adapts to heterospecific actions, showing hallmarks of social competence and cognition. These findings expand our current understanding of what is leadership and what is sociality. (L36-42)

R1.12. c) In the introduction, it might be helpful to mention explicitly, that the octopus is the main driver of the hunting group's formation and is therefore considered the nuclear species (see Mangini et al. 2023 *Philos Trans R Soc Lond B Biol Sci*). Or do the five fish species also form hunting groups without an octopus?

REPLY: Indeed, we have re-arranged this section to add this information and make the text clearer.

Octopus-fish(es) hunting groups have been mostly considered as "nuclear-follower" (or "producer-scrouter") systems^{10,25}, where the octopus is the nucleus of the group as it induces and maintains group cohesion²⁶. (L75-77)

Line specific comments:

R1.13. 1) Lines 149 – 153: Can you include a hypothesis on why blue goatfish are the main drivers of movement initiation?

REPLY: We added to this section the mobility and intraspecific collaborative hunting aspects typical of this species. Interestingly, this further reinforces the idea that the patterns seen are not driven only by mobility, as yellow goatfish (same species, different color phase) are equally mobile (Tables S29&33) but pull less frequently.

Goatfish, particularly blue and yellow (same species), are active predators with high mobility that find and corner prey together with other conspecifics³². These fish seem to use similar strategies in interspecific groups, serving as social cues for others. However, while blue goatfish have the highest pulling influence, equally mobile yellow goatfish (presumably a younger phase of the same species) exert less influence on the movement of others. This contrast between phases suggests changes in hunting performance, or potentially different strategies, with age³³.(L151-157)

R1.14. 2) Lines 209 – 211: Might it also be possible that the main function of the blacktip is to maintain group cohesion? Was web-over frequency increased in groups without blacktips? If so, this would indicate a negative influence of increased group cohesion on environmental exploration and consequently hunting success in groups with blacktips present. Instead, an increased group cohesion could also have positive influences on the group's success, e.g. by facilitating predation avoidance.

REPLY: We did consider this line of thought, however, first, we saw no evidence of antipredator behavior at a group level. Whenever a potential predator approached (e.g. giant barracuda, blacktip reef shark) the group would disband, and the octopus fled to the nearest den. This would happen regardless of the presence of blacktip groupers, and while we do understand that perhaps one is less likely to be found by a predator if the group moves less, this comes at the cost of not finding prey. As less environmental exploration is also related to increased aggression in the group (e.g., punching activity), we did not find significant potential benefits from the negative feedback provided by blacktip groupers. However, we acknowledge that there were not an enough number of these events to acquire quantitative data, and this may warrant further investigation in the future. Weover frequency decreased in groups without blacktip groupers (except when comparing to the octopus hunting alone), particularly so for short-term webovers, indicating a positive influence of group displacement and less cohesion. We address this point more in detail in reply to R1.17.

R1.16. 3) Line 291 – 292: The number of punching events is significantly lower when blue goatfish are present than when blue goatfish are absent. You conclude from this fact that the presence of blue goatfish stabilizes the social network. However, the reduction in punching probability when blue goatfish are present, can also be caused by the fact that blacktips were often absent at the same time when blue goatfish are present. In 67% (39 of 58) of the subgroups in which blue goatfish were present, blacktips were absent. This means that in the majority of subgroups with blue goatfish present, the main recipient of punching, the blacktip, was not there, which most likely affected the probability of punching more strongly than the presence of blue goatfish.

REPLY: Both effects (stemming from presence/absence of blacktip groupers and goatfish) are not mutually exclusive of course, but as the referee correctly points out, we had not addressed which seems to impact most the probability of punching. We now addressed this explicitly by analyzing the probability of occurrences depending on presence/absence of the extreme phenotypes in the group (blacktip groupers, blue goatfish, and both together; also Fig. 4E is now color-coded accordingly). The sample size here is relatively low due to the infrequent occurrences of such cases. Our finding may hint that the punching behavior of the octopus in groups containing both species is more similar to groups in which goatfish are present without the blacktips, although this needs further investigation ($p = 0.07$) (Table S41, Fig. S15). Therefore, we find that the goatfish's stabilizing effect may impact the decision of punching the most, albeit the difference is not pronounced. We added a sentence on that:

However, punching is significantly decreased by the presence of blue goatfish ($p = 0.0011$, Table S40, Fig. S14B), with a potentially larger effect than the blacktips' presence (Table S41, Fig. S15), indicating that the presence of the main puller species helps stabilize social interactions. (L281-284)

R1.17. 4) Lines 333 – 336: The negative slope between web-over duration and web-over frequency is likely caused by the natural interaction between feeding time and feeding frequency. An octopus that spends a long time feeding at one food source, is able to visit a lower number of food sources in a given time window than an octopus that stays at one food source only for a short time period. Might it be that with the help of the blue goatfish the octopus is able to find food sources of higher quality than when

foraging alone? Exploiting these high-quality food sources takes longer than exploiting low-quality food sources. Consequently, the octopus can only visit a low number of high-quality food sources within 100 seconds. If the increase in prey-finding opportunities due to an increased radius for environmental exploration when the blacktip is absent is causing the increase in feeding success in the octopus, the frequency of web-overs should increase with an increase in the number of potential food sources.

REPLY: As the referee states, the duration and frequency are related through feeding time and frequency. And indeed, the octopus can only visit a small number of high-quality sources within 100 seconds, as it takes time to consume whatever is in these high-quality sources. However, an important issue is that short webovers are associated with failed (prospective) feeding events, whereas long webovers are associated with the presence of food (as shown by results of our baited experiments in Fig 5B, see also controls indicating no significant effect of fish presence per se, in Table S44 and Fig. S18). Thus, as seen from natural foraging data in Fig. 5, while the octopus hunts alone, the frequency of webovers is very high, but with very small durations, indicating several failed attempts to find (octopuses primarily hunt by using their arms to 'taste' and detect prey under rubble or in crevices) and capture prey, i.e. what is known as speculative hunting. Thus, increasing the duration of webovers is more clearly associated with successful foraging events, while increasing frequency is more associated with foraging effort. But indeed, as the referee suggests, the frequency of webovers per unit time tends to decrease during successful foraging. So, when the referee asks "Might it be that with the help of the blue goatfish the octopus is able to find food sources of higher quality than when foraging alone?" – yes, our data suggest this is occurring. Indeed, we also agree that the presence of goatfish enables the octopus to restrict the number of potential food sources, selecting and moving only to high-quality ones. One could make the analogy to an 'extended sensorial system' (similar to Berdahl 2013 Science), which samples the environment in a much faster way than the octopus alone could. We thank the reviewer for this comment, and have added a few lines with this information

For the octopus, fish (in particular goatfish) appear to act as an 'extended sensorial system' (sensu 14) that samples larger spatial areas of the environment at a faster rate than the octopus could via direct sensing. This allows the octopus to filter the number of possible food sources via social information, seemingly saving energy (i.e. less web-over effort to find prey) and moving only towards high quality options (i.e., containing prey). (L339-343).

R1.18. 5) Lines 485-486: The word "of" is missing. Multi specific hunting groups can be composed of several species.

REPLY: Done

R1.19. 6) Line 487: Please write out the genus name for the blacktip grouper. Short names are not in bold, as stated in the brackets.

REPLY: Done, we were referring to the short names in bold presented in the figure. Changed for clarity.

Short names in bold, present in the figure, are used hereafter. (L789)

R1.20. 7) Fig. 2: I like the directed hierarchical social network plots. They convey an immense amount of information. However, I don't understand the meaning of the letters a, b, c. Significance between which parameters do they indicate? I further suggest to remove the coloring of the nodes. The same information that is conveyed by node color is also provided by the relative position of each species' picture (node) in the coordinate system. The different node colors are only distracting. Especially the octopus in Fig. 2B, 2C and 2D is hard to see because the node color is very dark. What is indicated by the asterisk in the box next to the octopus in 2C? And, what is meant with the error bar and asterisk in the middle of 2D? Shouldn't the red arrows in 2C actually be yellow, as they indicate frequency as in 2A and not efficiency as in 2B?

REPLY: a,b,c represent significant differences on the parameters defining each axis, which is why they usually have two colors. E.g., in Fig. 2a, yellow represents initiations, and blue represents pulling frequency, as per the figure caption. To clarify, we have now also colored the name of the parameter. Letters show pairwise significant differences as is customary in multiple comparison scenarios. Thus, e.g. species that have an *a* are different from *b* and *c*, but species with *ab* indicates that they are only different to *c*, and not to species that have the letter *a* or *b*. We have attempted to make it clearer, specifically in the caption of Fig. 2:

To illustrate multiple comparisons statistics, different letters between species indicate significant differences, and are colored according to the corresponding parameter in each axis. For example, a species labelled as *a* is significantly different from species labelled as *b* and *c*. However, a species labelled *ab* is only different to species labelled as *c*, and not with *a* and *b* labels. (L809-813)

We have kept the colors of the nodes as the purple asterisk in 2C indicates that there are also significant differences in the 'purple' dimension (given by the interplay of both xy parameters). That takes us to Figure 2D (via purple box), where the analyses are made explicitly on that purple dimension, and the significance is indicated with the asterisk for consistency. The 'error bar' was indeed deceiving, and we removed it. Yellow, red, and blue represent initiations, anchoring, and pulling, respectively, not frequencies or efficiencies.

R1.21. 8) Line 509: "Group composition" instead of "Species composition"?

REPLY: Done

R1.22. 9) Line 518: Remove "the" between "combined" and "factorial".

REPLY: Done

R1.23. 10) Fig. 4: The photos in A and B are meant to demonstrate dynamic events. Dynamic events are hard to see in a single image. It may help to provide a series of images instead. The labels "A" and "B"

21are almost not visible. Maybe changing the font color to white will help. Could you please indicate subgroups with and without blacktips in E by colored symbols? I guess that group composition, especially the presence of blacktips, in addition to centroid displacement is affecting the probability of punching.

REPLY: Indeed, we understand the reviewer's point but we tried not to make our figures too cluttered, so that the main message could be retrieved by glancing at them. However, we did add supplemental videos of these behavioral actions to make them clearer for the reader, and have now included a further one with fish displacements (Video S3). We also changed labels in Fig. 4 to white, as suggested. Also following this comment, and after producing Table S41 answering R1.16., we now colored symbols in Fig4e dependent on the factorial combination of presence/absence of extreme phenotypes.

R1.24. 11) Supp. Material Line 86: please explain what the reason was for dividing the hunting scenes into blocks of 100 seconds. How was that done, simply by cutting each scene after every 100 seconds, or were specific starting points defined for each 100-second block?

REPLY: We chose 100 seconds as it was the shortest duration of a hunting event (20181006-AM-I is made up of two blocks, where in-between the animals did not move for 5 minutes). After that, as there were no similar significant stops in other events, and to reduce any bias or subjectivity about choosing specific starting points, we indeed cut each scene every 100 seconds as the reviewer reasoned. We added this information in the Supplemental Material:

The smallest duration of a hunting event was 100 seconds, while the largest recording of a hunting event reached approximately 1800 seconds. Thus, to standardize this variability in duration, we used the smallest duration of 100 seconds, and divided all groups in 100 second blocks, totaling 107 subgroups. (L443-447)

R1.25. 12) Supp. Material Line 87: Please indicate of how many different octopus individuals the 107 subgroups consisted. Please also mention the average number of group members and the average group composition.

REPLY: As we filmed 13 different groups in several locations over different days, we approached the data as if these were 13 different octopuses. It is possible that the same octopus was filmed over different days, as occasionally we found octopuses hunting in close locations. It could also be that we filmed the same octopus on different locations after a change of den. However, there is no way of knowing with certainty. Unless distinctive features are present (such as the loss of an arm, or skin damage in a specific region) octopuses are virtually impossible to distinguish, especially in the field. This is further aggravated by the known tendency of octopuses to rotate through dens that were previously occupied by other octopuses (particularly in the same size class). Thus, we chose to assume that the 13 different groups were composed by 13 different octopus, while at the same time statistically controlling for hunt id, and thus potential octopus ID, using hierarchical random effects models to account for uncertainty, see Methods L603-609). This also minimizes potential strong claims on iterative interactions among the same octopus and fish individuals over extended periods of time. However, in future field seasons, we will

employ methodologies that will allow the individual identification of both octopuses and fishes, which will enable answering questions regarding long-term individual preferences. Average group composition is derivable from the values already present in the text, but we included information on average number of group members as requested:

The size of the groups varied between 2 and 10 individuals, with an average of 5.72 individuals, always with 1 octopus present. (L447-448)

R1.26.13) Supp. Material Table at Line 164: In the description of the Interpretation of Efficiency for Anchoring, “(probability of success)” should read “(probability of failure)”, shouldn’t it?

REPLY: This metric refers to an individual successfully anchoring another, i.e., the potential initiator came back (it is only a failure from the perspective of the initiator). We have moved this Table to the rest of the supplemental materials and introduced the following caption:

Table S48. Overview of the most important pull-anchor metrics, from the perspectives of the potential initiators (Pulling) and potential anchorers (Anchoring) of movement.

Reviewer #2 (Remarks to the Author):

R2.1. This is a valuable study. The authors have completed an excellent job at demonstrating the different roles within this unique multispecies hunting group. The authors test whether octopuses and various fish species drive leadership by using a pull-and-anchor analysis. Importantly, the authors include jittering interference by using a noise threshold of 0.1m. By adding this small amount of random variation to the data, the authors can reduce random fluctuations that might obscure the true patterns of leadership within the hunting group. The findings of this paper offer insight into how a typically solitary species, the octopus, can demonstrate significant behavioural flexibility within a social setting. This enhances our understanding of the ways in which the behaviour of solitary species can be influenced by social pressures in a heterospecific context. My concerns are minor and listed below. They primarily pertain to the sentence structure and language used in the manuscript. In several sections, the sentences are excessively lengthy and overly technical. I've provided suggestions aimed at enhancing the manuscript's readability and accessibility. While these concerns are not major, addressing them is necessary before I can endorse this work for publication.

REPLY: We thank the reviewer for their constructive criticism, and enthusiasm for the findings. We have tried to make the writing more straightforward and incorporated their suggestions in the text, particularly in the introduction and the last sections of the manuscript.

Main text

R2.2. Line 38: Would the term 'initiating' be more appropriate than 'inducing' here?

REPLY: Rephrased as 'stimulating', the direct opposite of inhibiting.

R2.3. Line 39: It's unclear to me whether you mean that 'classical leadership fails to accurately describe ALL complex heterogenous systems or just this one between the day octopus and different fish species. Please clarify in the text.

REPLY: We have rephrased to say that at least some complex heterogeneous systems can not be explained by the simple metrics of classical leadership (but keeping the sentence short due to word count).

Thus 'classical leadership' can be insufficient to describe complex heterogeneous systems (L36-37).

R2.4. Line 44-46: I'm getting a little lost here... can you rewrite please; also is the term 'same-species' necessary to include here?

REPLY: Simplified to: This seemingly non-social invertebrate flexibly adapts to heterospecific actions, showing hallmarks of social competence and cognition. (L40-41)

R2.5. Line 46: Please remove the term 'comprehensively', no need for it.

REPLY: Done

R2.6. Line 49: 'constituent components'... seems excessively complex and hard to read. Can't you just say participants instead? I would break down this entire sentence. For e.g. Collective behavior is the result of dynamic interactions between individual parts. It's vital for the coordinated functioning of various systems, whether they are physical, cellular, or social.

REPLY: Rephrased for simplicity as: Collective behavior emerges from the network of interactions among individual parts. It is central to coordinated functioning across scales of organization, including physical¹, cellular², and social^{3,4} systems. (L43-45)

R2.7. Line 52: what do you mean by 'informational differences'?

REPLY: Differences in the information that each individual has about the environment, we have rephrased. Heterogeneity, driven by genetic⁴⁻⁶, physiological^{6,7}, and informational^{4,5,8} differences among individuals, plays a vital role in explaining the functional complexity of collectives. (L45-47)

R2.8. Line 53-56: Similar comment as above. This sentence uses a lot of technical terms and is a lot to process. I would reframe. For e.g. We see this complex behaviour in actions such as the division of labour among cells and in insect colonies, as well as in how groups make decisions. These behaviours open up various leadership possibilities, from having a single leader to sharing leadership responsibilities among many.

REPLY: We have changed the text and thank the reviewer for the suggestion:

We see the emergence of such complex behaviors in the division of labor among cells² or insect societies³. The same functional dynamics also enable several alternatives of decision-making in groups, which can be placed over a despotic-democratic axis, i.e., one individual leader and shared leadership⁹⁻¹¹. (L47-50)

R2.9. Line 65: consider removing the term 'consequent'

REPLY: In this instance, we chose to keep this word, as it shows that one process (role specialization) is a direct consequence of the other (division of roles). Removing it would suggest that these processes could be independent or parallel, which would be misleading.

R2.10. Line 83: consider removing the term 'concurrently'

REPLY: Done

R2.11. Line 84: please reframe the last part of the sentence i.e., with suggestions that the octopus might either disregard the fish or be exploited by them.

REPLY: Done

with suggestions that the octopus might disregard²⁴, or is being exploited by, fish²⁸. (L80-81)

R2.12. Line 113: would the term 'efficacy' (instead of efficiency) be better here, as I believe you're interested in the ability to produce a desired or intended result?

REPLY: While it is true that 'efficacy' could be used to describe the ability of achieving a result, here we were interested in weighing the result by the effort required in achieving it. Therefore, we feel that efficiency is a better term.

R2.13. Line 119-126: Really nice explanation of the 'pull' and 'anchor' process.

REPLY: Thank you

R2.14. Line 156-160: Please divide this into two sentences. Perhaps start a new sentence after 'movement in others'

REPLY: Done:

However, while blue goatfish have the highest pulling influence, equally mobile yellow goatfish (presumably a younger phase of the same species) exert less influence on the movement of others. This contrast between phases suggests changes in hunting performance, or potentially different strategies, with age³³. (L154-157)

R2.15. Line 190-194: Consider breaking these two sentences in two. For e.g., the current findings reveal that groups made up of individuals from different species, which are not related and have evolved differently, can achieve complex functions. These functions are similar to those qualitatively described in groups of related individuals from the same species, who exhibit high levels of social interaction.

REPLY: Rephrased:

In these systems, lower-ranked initiators comprised mostly of younger males, would provide different direction options for older males to choose from, the latter acting as the deciders in the troop. Thus, multispecies groups composed by differently-specialized, non-related, individuals can achieve the same functionality dynamics and coordination shown by highly-social species.(L190-194)

R2.16. Lines 229-232: Odd that you only report octopus pulling success here, what about the aforementioned barbel goatfish?

REPLY: Changes to barbel goatfish pulling efficiency were non-significant, and thus not explicitly referred to in the text, also due to word count limitations. However, the information is present in Fig. 3c and Table S23.

R2.17. Lines 240-246: Can you please reword; I get lost here.

REPLY: Rephrased:

However, by explicitly analyzing groups according to the presence of extreme phenotypes — i) with blue goatfish (blacktips absent), ii) with blacktips (blue goatfish absent), iii) with both species (Fig. 3D) —, we found that groups containing blacktips become less mobile and more compact around the centroid ($p < 0.05$ in comparison to blue goatfish only groups, Table S26-27, Fig. 3D). (L237-240)

R2.18. Lines 289-291: Such a great finding!

REPLY: We agree! Moreover, after suggestions made by R1, we also provide further evidence that the presence of blue goatfish is stronger than the presence of blacktip groupers in modulating punching (Table S41 and Fig S15).

R2.19. Lines 300-304: long sentence, please trim down. You can remove 'in order' for instance and remove terms like 'considerable'.

REPLY: Rephrased:

To understand the functional consequences of these species-dependent modifications to the system's properties, we can ask how foraging strategies and benefits for the parties involved are impacted. (L293-295)

R2.20. Lines 313-318: Clever!

REPLY: Thank you

R2.21. Line 333-334: 'Indicating' used twice in quick succession.

REPLY: Done.

R2.22. Line 340-343: This hurts to read. Try this: In terms of ecological functionality, combining the identified patterns of movement influence and the changes in foraging strategy and benefits reveals

specialized hunting roles within the group. The goatfish explore and identify various prey locations, thereby providing options, while the octopus predominantly makes the decision on which options to pursue and attempts to capture the prey. These joint actions result in higher prey capture success rates.

REPLY: This is a welcome simplification of the previous text, thank you. We have changed it accordingly:

Combining the identified patterns of movement influence and foraging strategies reveals specialized hunting roles within the group: goatfish explore space finding different prey locations, thereby providing options, while the octopus predominantly decides which option to take and attempts to capture the prey. Together, these joint actions can yield higher prey capture success rates for the octopus (this study) and for fish^(21,27,28). (L333-337)

R2.23. Lines 347-348: This is really interesting, could you elaborate or provide an example of where we might see this in the animal kingdom (outside of the octopus-fish hunting interaction)?

REPLY: We are honestly unaware of a well-studied example providing similar conclusions. For instance, cleaner fish have very complex associations with heterospecifics, but also have a complex intraspecific social life. Another example of complex joint hunting could be the coyote-honey badger system, but to our knowledge, these reports are based on quick observations from camera traps and short videos, and no behavioral flexibility from the honey badger (the otherwise-solitary species) has been reported. We also briefly discussed the moray eel-grouper system. We have elaborated on this topic in the discussion, and thank the reviewer for prompting this reasoning line:

This functional complexity and its dynamic nature appear to distinguish this system from what is known of other interspecific hunting associations, such as badger-coyote³⁹ or mixed-species bird flocks systems^{12,40} (albeit these are understudied systems), as well as the well-studied pairwise moray eel-grouper system¹⁹, where partner control mechanisms seem absent. The exhibited range of partner-dependent behavioral flexibility, especially concerning the use of social information when deciding to switch foraging strategies and whom/when to punch, indicates that day octopuses have hallmarks of (heterospecific) social competence and cognition. Such ability to flexibly respond to non-predatory heterospecific actions within a complex dynamical system is rare (or underreported) in animals. One particularly known example are cleaning stations. Here, iterative interactions between resident clients and cleaner wrasses make clients rely on dynamic positive (cooperation) and negative (cheating, punishment) feedback mechanisms to choose among stations to visit, which conversely changes the behavior of the cleaners themselves, leading to the emergence of a “biological market”⁴¹. However, most animals that develop complex interactions with heterospecifics share complex social lives with conspecifics *a priori*, in principle transferring similar decision-making rules to heterospecific scenarios⁴². Day octopuses, often considered solitary or at least asocial with conspecifics, can have a sophisticated cross-species social life without prior interactions with conspecifics, which, to our knowledge, is unique in invertebrates and extremely rare in the animal kingdom. (L347-364)

R2.24. Lines 349: isn't it closer to 550 mya?

28REPLY: Indeed, updated.

R2.25. Line 359: I think you can drop the word comprehensively.

REPLY: Done

R2.26. Line 496: It might be my screen but the colour looks more like purple rather than blue.

REPLY: We had used the default 'blue' provided by the package used, but we agree that it was 'leaning' a bit towards the purple and have changed it.

Supplementary File

R2.27. Line 87: A little confused here, how was there a total of 107 subgroups but within that, you've created 130 blacktips sequences? Were the subgroups divided further?

REPLY: In each subgroup, several numbers of blacktips could be present, i.e., between 0 and 4. Therefore, in some cases, we sampled more individuals than the number of subgroups. Subgroup species and individual composition is described in Data S1.

R2.28. Line 125: Great and important that you included this.

REPLY: Thank you

Reviewer #3 (Remarks to the Author):

R3.1. The key contribution this paper is claiming is that three components of leadership on movement decisions (where, when, and if, to move) can be split across different individuals within a collectively moving group. This is a really exciting and important suggestion, and the authors have gone to enormous analytical effort to provide supporting evidence for it. Unfortunately they are slightly let down by the writing in the paper: it is needlessly obfuscating in many parts, to the point of being distractingly so in many places, which did make it hard to review and assess. I do hope they make some effort to improve clarity in many places as it is hard enough to understand for native English speakers, let alone non-English speakers. After reading this paper thoroughly, I'm afraid that I do have two main issues that do need clarification/additional analyses to resolve.

REPLY: Thank you for taking the time to provide feedback on our manuscript. We have made substantial edits to the text in response to these comments and the comments of the other reviewers, and hope that the paper is now clearer.

R3.2. Firstly, one of my main questions concerns the 'pull-and-anchor' metric used by the authors, which was made initially tricky to understand because I cannot find any reference to the sample size of dyads used in their analysis (the supplementary data files referred to are not included in the reviewer documents).

REPLY: We are sorry about the missing supplementary data files. We provided those on dryad and zenodo, and referred to this in the manuscript text and the reporting summary, so we think we were not at fault. The supplementary contains some important information which could have helped to solve some of the issues raised by the reviewer. But perhaps there were some issues during the manuscript transfer. The reviewer is correct in the assumption that dyads are indeed made between every individual in the group, however they do not necessarily provide a non-zero output (see reply to R3.3).

Below we paste the co-occurrence matrixes that provide the dyadic sample sizes for the whole dataset (SCode 3.pull-anchor, L385-391).

#	blacktipgrouper	barbelgoatfish	lyretailgrouper	octopus	bluegoatfish	yellowgoatfish
# blacktipgrouper	72	166	20	132	47	20
# barbelgoatfish	166	27	28	95	47	8
# lyretailgrouper	20	28	12	42	76	9
# octopus	132	95	42	0	102	23
# bluegoatfish	47	47	76	102	44	43
# yellowgoatfish	20	8	9	23	43	4

Table S47 now contains this information.

R3.3. Specifically, it wasn't clear how the authors chose each dyad for each event when calculating this metric, particularly individual 'j' (the individual that responds to an initiator), and most importantly when this individual was classified as an "anchor". We have to assume that the authors are copying this component of the methodology used by Strandburg-Peshkin & Farine et al (2015), which they reference under the caveat of making several changes, where for each 'event' the authors created dyads between the single 'initiator' and every other individual within the social group. I find this problematic since the language used by the authors throughout the paper (e.g. "initiation attempt", "pulling frequency" etc), and their analyses, makes the implicit assumption that a dyad constitutes a causal event where a social decision to follow or not is being made between two specific individuals. However, in the majority of 'events', this is not actually true since the information about the 'true dyad' (i.e. the specific individual/individuals an animal is choosing to associate with) will be swamped by the data generated from every other possible connection between the initiator and all other group members. There is little intuitive predictions that can be made about how this will affect the author's results, except that it will likely over-inflate the calculated importance of the non 'true' leaders.

REPLY: For identifying pull and anchor events, only interactions where individuals register meaningful changes in relative distance are registered (consistent with the approach used in Strandburg-Peshkin et al. 2015). The way these are calculated and the thresholds applied (concerning noise reduction, disparity of movements, strength of interaction) were referred to and explained in the methods section (now L482-499). Importantly, this method does not result in every individual being considered as involved in every event – only those that show meaningful changes in dyadic distance are included. Thus, for example, if one individual simply closely associates with another all the time, there will be no pull-anchor interaction between the two. It is true that the initial version of our manuscript did only included analyses that operate at a dyadic level, and that the metrics we derive from these "events" represent aggregations across these dyadic events. As a similar question was raised by R1, we performed further analyses to remedy this issue. Please check our response to R1.5 (but also R1.4 and R1.3).

As can be seen there, our conclusions are highly robust to the exact way influence is estimated, including moving away from a purely dyadic framework. Specifically, in one of the additional new analyses, we looked at only 1st pullers for any specific follower in a given (multi-individual) event. In another, we looked at group-level pulls (between one individual and the relative centroid of the group). These analyses show that the results we found are not artefacts of the dyadic approach (see R1.3-5, Fig S3-5 and Tables S11-13 & S15). Regarding the nomenclature and methodological principles used, they are the same as in the Strandburg-Peshkin et al paper. While 'attempt' can perhaps be too inferential (we changed the nomenclature to simply 'initiator' or 'initiations'), initiators, pullers, and anchors, is the standard nomenclature. They characterize the physical process and the outcome, not specific intentions, as in Strandburg-Peshkin. However, as referred in that paper, while observational studies do not capture causality explicitly, it is statistically improbable that all individuals in a group would make the same causally independent decision (despite considerable individual differences), at the same time, across hundreds of such decision events, as would be required to explain group cohesion for the duration of hunting events. We do concede of course that some level of independent decision-making forms some part of the dynamics we observe. However, we have shown that the patterns obtained are consistent for not only all meaningful dyads, but also 1st initiators- and centroid-levels analyses, and are further

31supported by analyses on preferential directions and deviation from previous paths (see R1.8), which allows us to state with some confidence that these independent decision-making events become non-relevant to our conclusions. Of course, such still disallows conclusions regarding 'intentions' for all cases, but it does allow us to affirm general patterns of animals pulling/anchoring others, and initiations (particularly for the ones that result in pulling). We will address further points below, where the reviewer also expanded on them. Nevertheless, to address the legitimate concerns of the reviewer and highlight that it is not our goal to infer intentions in every case, but measure the process and physical outcome, we have added the following regarding nomenclature and the correlation/causality issue in the methods:

As referred in ¹¹, while observational field data do not explicitly capture causality, it is statistically improbable that all individuals in a group would make the same causally independent decision (despite considerable individual differences), at the same time, across hundreds of decision events, as would be required to explain group cohesion for the duration of hunting events. While some level of independent decision-making forms some part of the dynamics we observe, we have taken several steps beyond data thresholding to ensure that statistical patterns found are consistent, allowing us to state with some confidence that these independent decision-making events become inconsequential to our conclusions.

First, to ensure that individuals were not simply moving randomly in space, we retrieved the angles between the relative movement vectors of the initiator (during $t_1 - t_2$) and the potential follower (during $t_2 - t_3$) when pulls and anchors occurred (Fig S19, see also see 1.general script). First, we performed a Rayleigh's test and verified that the frequency distribution is non-uniform, i.e. there is preference for movement in certain angle ranges ($p < 0.0001$). From plotting a histogram of the most frequent pull and anchor angles in our dataset (Fig. S19), a clear difference in probability emerge, statistically different from what one would expect from 360 degrees random movement (i.e. movement equally likely to happen at any given angle). In our system, anchors are actually 2.3667 times more likely than pulls, if an individual would move in a random angle relatively to another. Moreover, if we calculate the probability of pulling or anchoring for each given angle (rounded as integers between 0 and 359), we can then obtain the average of all probabilities across integers, to get the mean probability of a pull or an anchor occurring at chance level. In our system, the mean probability of pulls is around ~31.31% (median probability is even lower: 20%) and of anchors is ~68.69% (median probability: 80%) (see end of 1.general.ipnyb). Noteworthy as well, the narrow angle range at which pulls occur (relative to any other angle), show a bimodal high frequency distribution around 30 and -30 degrees, with a small number of pulls occurring near 0 degrees (Fig. S19). Thus, in the large majority of cases, pulling does not occur due to movement inertia — i.e. individual j apparently follows i but such happens because j was moving in the same direction —, individuals deviate from their course and actively follow others. (L525-551)

R3.4. The authors seem to (ambiguously) imply this on lines 164-167, where they suggest the strong result showing blacktips are highly efficient pullers and anchorers might be caused by a "signal-to-noise ratio" issue. Beyond being completely unclear what this actually means (they really need to change the language here) the only reason they seem to suggest this is because this result did not follow their predictions. Indeed, I assume that the explanation the authors are attempting to make is that blacktips are being classified as highly efficient anchorers and pullers purely because they are associating more strongly with the octopuses than with any other fish, and the other fish within the social group are actually

32only (or almost entirely) paying attention to the octopuses when deciding whether to return after an “initiation attempt”. This issue with the non-specific classification of social relationships, coupled with the correlational nature of the study, do make me question how much of the other main results are also artefacts. I don’t think this was an issue in Strandberg-Peshkin’s study since they tested directly whether following events were being driven by single vs multiple individuals, but I am not really sure how this issue can be resolved here: to really assess the bias created from this metric, some form of simulation study will be needed where the metrics calculated for true leaders can be compared to that of individuals that simply associate very strongly with leaders. If the authors cannot validate this metric in this manner, then they must explicitly state the shortcomings of this metric so a reader is made aware of its issues, and to change the language used throughout the study to something more appropriate for its purely correlational nature.

REPLY: We apologise for the ambiguity with respect to the signal to noise. What we suggested was that the blacktip groupers rarely move. Thus, when they do (or they do not, while others do), may be perceived as especially salient (i.e. higher signal-to-noise ratio). This follows the terminology employed in neuroscience and detection theory, where individuals compare the ratio of relevant information-bearing patterns (signals) to random patterns (noise). However, the referee is correct that this is just a possible explanation for why the blacktip grouper may be an effective puller/anchorer. We have now changed the text to make it clearer that this is deliberately speculative regarding the reason, but we still think it worth noting as it may be useful to guide future studies of such phenomena where this relationship could become a focus. Since it is not critical to the study, we could also remove this if there remains strong objection to such speculation. We also agree with the referee that further analysis was necessary to address the possibility of indirect effects. To do so, following the suggestion R1 made, we have now additionally conducted analyses on only the 1st pullers (thus eliminating ‘byproduct’ pulls’ from simply associating strongly with 1st pullers), or centroid level-pullers, and confirm that these results are highly congruent with the original analyses. Please see comment above regarding nomenclature. We changed this section to:

We anticipated that blacktips, as they rely more on ambush predation, would exclusively follow others. This was not the case. Their unexpected status as highly efficient pullers and anchorers may be due to a higher signal-to-noise ratio in movement (following signal detection theory, as in ³⁴), comparatively to other species. Since blacktips are ambush predators, they spend longer periods relatively immobile than other hunting partners (moving less than 1 cm/s; Tukey HSD, $p < 0.05$, Table S14, Fig. S6). Therefore, initiation (or lack) of movement on their part may provide a clearer and more salient (greater “signal to noise”, using the terminology of signal detection theory), cue to other group members that prey may or may not be nearby, serving as ‘quality’ indicators. (L164-172)

R3.5. Secondly, this brings me onto the octopuses, and whether they are indeed following other fish to the extent that the author’s conclusions in the abstract “specific fish species (particularly goatfish) drive environmental exploration, deciding where [.....] the group moves” are valid. The major limitation with this claim is that the efficiency by which octopuses do actually get ‘pulled’ by goatfish is so low, as is the goatfish’s ‘general influence’ on collective movement, that I don’t find this nearly convincing enough to

justify the claims made in the paper about leadership being multidimensional. When assessed in concert with the ambiguity of what the pulling/anchoring metrics are telling us, it is conceivable that these results could be obtained if goatfish use the same information as octopuses about where to forage (e.g. the quality of foraging sites via olfactory or visual means), and are just more likely to be at these before an octopus makes its own independent decision, due to differences in mobility and tendency to practice their own independent foraging relative to other fish. The key result needed here is whether octopuses use social information in preference to personal information about where to go when there are multiple equally-suitable options to forage. Beyond an experiment in a standardised foraging arena, I wonder if the authors can provide something approximating this analysis by using their 3D scenery reconstructions to a-priori objectively identify potential foraging sites (e.g. large rocks)? If so, it would be very convincing (to me) if they can then directly test the importance of various environment features between multiple potential foraging sites (e.g. rock size, adjacency, deviation from previous path angle etc) versus social information (if and what species of fish are foraging there).

REPLY: As discussed in response to another reviewer comment, the efficiency that octopuses get pulled by goatfish is actually not low, and this misunderstanding results from the (reasonable but mistaken) assumption that the expected proportions should be 50% pull and 50% anchor (please see the reply to R1.8 and R3.3 above, where we detail a directional analyses showing that these are not random events). Not only the mean probability of pulling is only around 30% if one were to assume random movement relative to others, but also that individuals deliberately change their direction of travel to follow others.

The reviewer raises an interesting point about obtaining more evidence for an explicitly social factor being employed. Further to this, we realized—thanks to the comments of the referee—that we can ask this question of our field experiments in which our structures were attacked by fish previously to the octopus moving closer to it. Moreover, we have also re-analyzed the data from the field experiment considering events where the octopus did not engage with the structure, as well as events where the structure was previously attacked by fish or not (all information in Data SX). First, we show that structures had ~50% probability of being attacked by octopuses (in this case, given that a novel object is being introduced in the environment, 50% probability of interacting with it or not could be said to be random), regardless of having food or not (Fig S17, Table S43). However, when these structures were previously attacked by fish, the probability of an octopus attacking increases to ~90% (Fig S17). Thus, as may be expected in this type of marine environment where visual information usually cannot indicate prey location and relatively far olfactory cues are subject to considerable turbulence, octopus would tend not to have reliable personal information regarding relatively distant features in the environment. However, social cues, by contrast, are very much taken into account when foraging (L301-311). In addition to providing a more direct test of the hypothesis, other methods, such as analysis of the 3D structure, are biased by the fact that we only record the structure local to where the octopus does go, as opposed to additionally where it could, but does not, go. This would complicate interpretation of such a relatively indirect evaluation of whether the octopus employs social information. While they could be interesting to explore in future work combined with other metrics or experimental manipulations, we see them as beyond the scope of this already long and complex manuscript. We have made the following modifications to the paper:

To experimentally validate the observational parameters, and remove possible confounds, we performed a field experiment where empty and food-baited (i.e. full) structures were placed near the octopus (Fig. S16). We found that octopuses were equally likely to perform web-overs on empty and full structures, but nearly always (i.e. 95% of the times) performed them over structures that were previously attacked by fish ($n = 113$, $p = 0.004$, Table S43, Fig. S17, Video S5). Moreover, we found that web-over duration was independent of fish presence (Table S44, Fig. S18), but that web-overs on full structures were longer than those on empty structures ($n = 43$, $p < 0.0001$, Table S44, Fig. 5B). Thus, octopuses actively incorporate social information from fish when making prey-oriented decisions, and increases in web-over duration are tightly linked to successful food acquisition/prey capture, not, e.g., potential kleptoparasitism avoidance. (L303-313)

Field experiments

Complementary field experiments were conducted at Lizard Island Research Station in Australia, where baited (prawn *Penaeus monodon*) or empty U-PVC tee-fitting structures were placed ~50 cm from octopuses, to gauge how web-over characteristics, i.e. occurrence (interaction with the structure or not) and duration of web-overs, were impacted by: i) presence of food, ii) previous strikes on the structure by fish, and iii) presence of fish (Fig. S16, Video S5). We used 5 minutes (300 seconds) of web-over duration as maximum trial time, as this meant that the octopus had taken the structure back to its den (in these cases, web-over duration was not considered). This experiment was performed under Permit G23/47925.1 from Great Barrier Reef Marine Parks Authority. (L396-405)

Minor comments

R3.6. Much of the language used to describe behaviours is inappropriately leading. For example, there is no evidence that the movements by any of these animals are “initiation attempts” (which implies intent and function): it’s much more parsimonious to explain this behaviour as animals moving off to forage nearby and then returning if they are followed: there is no need to infer their ‘intentions’, so please replace throughout.

REPLY: We agree that “initiation attempts” sounds too intentional and have made the requested changes by removing 'attempts' as explained in the above reply to R3.3 .

R3.7. Line 46: “comprehensively broaden our understanding of sociality, specialization, self-organization, problem-solving, control theory, and leadership, in collectives across biological scales”. I appreciate the author’s enthusiasm for their work, but this is so overtly-aggrandising that I suggest they change it to a more appropriate and level-headed statement to avoid putting readers off.

REPLY: We agree. We have rephrased this to focus on more specific aspects:

Thus ‘classical leadership’ can be insufficient to describe complex heterogeneous systems, where leadership instead can be driven by both stimulating and inhibiting movement. Furthermore, group

composition altered individual investment and collective action, triggering partner control mechanisms (i.e., punching) and benefits for the de facto leader, the octopus. This seemingly non-social invertebrate flexibly adapts to heterospecific actions, showing hallmarks of social competence and cognition. These findings expand our current understanding of what is leadership and what is sociality. (L36-L42)

R3.8. Line 196-199: here the authors are comparing observational differences in network structure between groups of animals as being akin to gene knockout studies. Again, there is no need for this level of grandiosity: please remove. In addition, I am familiar with the social network literature, and the term “to highlight how node removal can lead to network structure rearrangement” suggests that nodes were actually removed (i.e. it is an experimental removal study: I had to check the supplementary material to doublecheck this is false), when in fact there was no such thing. This part of the study needs to be clearly communicated as purely observational.

REPLY: Thank you for pointing this out. We have made the suggested changes.

To assess how extreme phenotypes impact individual and group-level properties, we conducted directed network and pull-anchor analyses on groups where these phenotypes were not present (Fig. 3A-C). (L197-199)

R3.9. Line 212: Where is the evidence that blue goatfish “play a key role in both group mobility (considerably impacting movement decisions)”? They do have an important role in pulling frequency, but their overall “general influence” is not significantly different to other species according to Fig 2D.

REPLY: Mobility explicitly refers to instances where there is movement stimulation. We think it’s clear that if blue goatfish pull other partners frequently, then they play a key role in creating movement in the group. See also analyses for pulling frequencies of the group’s centroid. And the results regarding group displacement that are presented a little after in the text. The general influence of blue goatfish is not different from other species because they do not anchor others as much.

R3.10. Line 266-300: I appreciate how interesting this behaviour is, but it does seem to come out of nowhere, and doesn’t seem to be contributing to the key messages made by this paper. Would it be more appropriate being written up as a separate manuscript on its own (to avoid being lost in the main points of this manuscript)?

REPLY: While we agree that this topic could (and should) be expanded upon further in future work, we have decided to keep it because we believe this is an important aspect to explain leadership, group properties, and the octopus’ behavioral flexibility.

R3.11. Line 300 onwards: a potential confound of this analysis is if octopuses are also more likely to web-over to avoid their captured prey being stolen, i.e. it is primarily used for capturing prey, but extended periods of webbing are also used to avoid already-immobilised prey that are being consumed from being stolen. I base this entirely on my own anecdotal observations that octopuses can sometimes move

around seemingly-normally whilst in the process of consuming a captured prey item (implying that webbing-over is not purely associated with prey handling and consumption). Can you authors please give a compelling reason why this cannot explain the fact that extended webbing over events occur when there are potential kleptoparasites around?

REPLY: First, if the octopus was simply preventing kleptoparasitism, then a pronounced increase of web-over duration should also be registered when only blacktip groupers are present. In fact, one would expect it in those situations even more so, as blacktip groupers are ambush predator, exploit others' movement, and are the most punched. Lastly, and also in continuation to the reply to R3.5, the re-analysis of the field experiment showed that octopuses do not change their web-over duration depending on the presence of fish (Table S44, Fig S18), only on the presence of food. We thank the reviewer for the comment, and inform that this information is now part of the text:

Thus, octopuses actively incorporate social information from fish when making prey-oriented decisions, and increases in web-over duration are tightly linked to successful food acquisition/prey capture, not, e.g., potential kleptoparasitism avoidance. (L310-313)

R3.12. There is no Data S1 file provided for reviewers to assess.

REPLY: We are unsure why the reviewer was not able to access it, data were made available on dryad and zenodo.

Decision Letter, first revision:

29th May 2024

Dear Dr Sampaio,

Your manuscript entitled "Multidimensional influence drives leadership and composition-dependent success in octopus-fish hunting groups" has now been seen by three of the original reviewers, whose comments are attached. The reviewers have raised a number of concerns which will need to be addressed before we can offer publication in Nature Ecology & Evolution. We will therefore need to see your responses to the criticisms raised and to some editorial concerns, along with a revised manuscript, before we can reach a final decision regarding publication.

You'll see that reviewers 2 and 3 are very pleased with the progress you've made and sign off, but reviewer 1 still has some remaining comments though these should be straightforward to address. Most importantly, they ask why the re-analysis they requested has been included in the SI only,

37rather than used as the basis for main text figures and discussion--please could you revise accordingly.

We therefore invite you to revise your manuscript taking into account all reviewer and editor comments. Please highlight all changes in the manuscript text file [OPTIONAL: in Microsoft Word format].

- * Include a "Response to reviewers" document detailing, point-by-point, how you addressed each reviewer comment. If no action was taken to address a point, you must provide a compelling argument. This response will be sent back to the reviewers along with the revised manuscript.
- * If you have not done so already please begin to revise your manuscript so that it conforms to our Article format instructions at <http://www.nature.com/natecolevol/info/final-submission>. Refer also to any guidelines provided in this letter.
- * Include a revised version of any required reporting checklist. It will be available to referees (and, potentially, statisticians) to aid in their evaluation if the manuscript goes back for peer review. A revised checklist is essential for re-review of the paper.

[REDACTED]

Nature Ecology & Evolution is committed to improving transparency in authorship. As part of our efforts in this direction, we are now requesting that all authors identified as 'corresponding author' on published papers create and link their Open Researcher and Contributor Identifier (ORCID) with their account on the Manuscript Tracking System (MTS), prior to acceptance. ORCID helps the scientific community achieve unambiguous attribution of all scholarly contributions. You can create and link your ORCID from the home page of the MTS by clicking on 'Modify my Springer Nature account'. For more information please visit please visit www.springernature.com/orcid.

38Please do not hesitate to contact me if you have any questions or would like to discuss these revisions further.

[REDACTED]

Reviewer expertise:

as before

Reviewers' comments:

Reviewer #1 (Remarks to the Author):

I thank the authors for carefully and comprehensively addressing my and the other reviewers' concerns regarding their manuscript. I was very happy to read that the analysis method I suggested produced results that reflected the original findings. However, I was very surprised to find all the figures in the main text to still be based on the old, invalid analysis method. Presenting the results from the new analysis only in the supplementary material as facts that support the original conclusions drawn from the old, invalid analysis is not sufficient. Why not using a valid method right away instead of arguing post-hoc that a valid method produces the same results as the invalid method? All results and figures presented in the main text need to be based on either a dataset of dyadic interactions in which all same-species interactions and all 2nd, 3rd, etc. order events are excluded (i.e. only the 1st pull in a sequence of pulling events in dyads consisting of different species is considered), or on a data set of interactions between a focal animal and the centroid of the rest of the group (my personal preference).

Line 314 – 332: Here, data sets from two different octopus populations (Eilat and El Qeseir) are compared. Please add to this paragraph one sentence clarifying to the reader that differences in web-over characteristics between solitary and group hunting octopuses may also be due to general behavioral differences between the two populations studied.

Line 444 – 447: I still don't understand why it was necessary to divide the hunting events into blocks of 100 seconds. Does that mean that a hunting event with a duration of more than 100 seconds was cut and considered as two separate events? When the authors are able to calculate the duration of hunting events ("The smallest duration of a hunting event was 100 seconds, ...") they must be able to define precise start and end points for such events. Wouldn't it be more correct and less confusing to clearly define hunting events and restrict analysis to these events rather than just chopping the whole footage into 100-second long chunks irrespective of when in these chunks a hunting event took place? How was the term "hunting event" actually defined? As a certain period of time before and after a web-over? How was the relationship between hunting events and pull/anchor events? Were only

39pull/anchor events analyzed that happened temporally close to web-overs, or were all pull/anchor events that happened within the 100-second block analyzed? If the latter was the case, and pull/anchor events were independent from web-overs, why fragmenting the footage at all? In addition, this paragraph still does not give a good overview of the actual sample size. The number of individuals given in line 449 – 450 is misleading because it is artificially enlarged by chopping up the footage into 100-second long chunks. I know it is hard to say from footage taken from wild animals, but I think it is important to know how many different octopus individuals were involved in this study to judge the generalizability of the results amongst *Octopus cyanea*. Otherwise, the limitation on generalizability of conclusions should be mentioned in the manuscript.

Line 805 – 808: Please add to the figure caption the meaning of the red tracks in the middle panel of Fig. 1B and the color-coded tracks in the bottom panel of Fig. 1B.

Line 861 – 863: Please also add to the figure caption that here the behavior of two different populations of octopus are compared.

Fig. 2B: Error bars of octopus and yellow goatfish are truncated at the y-axis. The same applies to the blue goatfish in Fig. 2C.

Fig. 2C: Please add to the figure caption the meaning of the asterisk next to the octopus in Fig. 2C. If the asterisk relates to Fig. 2D, it should be removed, since Fig. 2D is not a cutout of Fig. 2C.

Fig. 3B: Vertical error bar of barbel goatfish is truncated at the top.

Fig. 3D: In the bottom right, the small picture of blue goatfish is on the top and of the blacktip at the bottom for the first two box plots but changed for the last boxplot. The order of species should be consistent to avoid confusion.

Fig. 4A and B: I still don't think that still images are useful when one aims at displaying dynamic events. There is sufficient space in this figure to replace the two still images by two series of images taken at different times throughout the specific partner control behavior. The size of the bar plot in panel D can be reduced by 50%.

Fig. 4C: The dashed box around the octopus is not necessary. Error bars are missing for all species. Please indicate in the axis labels which axis is represented by the green and yellow letters that indicate significant differences.

Fig. 4D: Please remove the dashed box. The y-axis label should read "Proportion of received punches".

Fig. 4E: Replace upper case by lower case letter "D" in the word displacement in the x-axis label.

Fig. 5C: What is meant by "Web-over frequency (per block of 100 s)", the number of web-overs within 100 seconds? If so, it might be better to write "Number of web-overs in 100 s" or to calculate an actual frequency (events/second) value. As in Fig. 3D, it might help to keep the position of specific fish species constant (blue goatfish on top/blacktip at the bottom) in the pictogram next to the box

40plots in the upper right corner of the panel.

Reviewer #2 (Remarks to the Author):

Multidimensional influence drives leadership and composition-dependent success in octopus-fish hunting groups

Thank you to the authors for thoughtfully incorporating my feedback and making the necessary revisions. The paper now reads smoothly, greatly enhancing the clarity and accessibility of the science. I have a few additional minor suggestions outlined below. Overall, congratulations on an excellent study

Main text

Line 317: I realise there are multiple fish species involved but I think for readability, keep the term as 'octopus-fish' rather than 'octopus-fishes', you also use the term 'octopus-fish' below so best to stay consistent.

Line 319: For consistency, it would be worthwhile changing the term 'induces' to 'stimulates' as you've done above.

Line 329: Consider replacing the term 'conjectural' with 'speculative'.

Line 1978: remove the term 'predominately' or 'highly', you don't need both here.

Reviewer #3 (Remarks to the Author):

The authors have done a superb job at addressing my concerns (much more thoroughly than I had anticipated), and combined with their addressing of the other reviewer's comments, has allayed the few concerns I had about the veracity of their conclusions based on the analyses. Congratulations on a really exciting and novel paper.

*****END*****

Author Rebuttal, first revision:

Response Letter

Reviewer #1

41R1.1. I thank the authors for carefully and comprehensively addressing my and the other reviewers' concerns regarding their manuscript. I was very happy to read that the analysis method I suggested produced results that reflected the original findings. However, I was very surprised to find all the figures in the main text to still be based on the old, invalid analysis method. Presenting the results from the new analysis only in the supplementary material as facts that support the original conclusions drawn from the old, invalid analysis is not sufficient. Why not using a valid method right away instead of arguing post-hoc that a valid method produces the same results as the invalid method? All results and figures presented in the main text need to be based on either a dataset of dyadic interactions in which all same-species interactions and all 2nd, 3rd, etc. order events are excluded (i.e. only the 1st pull in a sequence of pulling events in dyads consisting of different species is considered), or on a data set of interactions between a focal animal and the centroid of the rest of the group (my personal preference).

REPLY: We thank the reviewer for their appraisal and comments, which have greatly benefited our manuscript. While we had presented data supportive of our conclusions, it was indeed paramount to show that these conclusions were maintained when subsetting the data, e.g., just to consider 1st pullers or removing same-species interactions. This was important to make sure that our conclusions were not the result of hidden artifacts or biases within the dataset, stemming from differences in group composition and total number of individuals, or number of individuals present of the same species. We respectfully disagree however, that this makes our initial method invalid. To the contrary, in our view, these analyses are complimentary, both informing us about the system dynamics. Furthermore, the congruence in findings supports the validity of the original method when used in a large-scale dataset, by reproducing the same results. Moreover, our initial analyses allow for a more nuanced understanding of our dataset, e.g., by showing which species pull more other species (while still providing a general pulling 'score'), and the directionality of influence in species-level dyads.

Case in point, the hierarchical networks, as the reviewer pointed out before, convey an immense amount of information in an organized way, both for the general group dynamics (Fig. 2) and for depicting changes in dyadic interactions driven by group composition effects (Fig. 3). A purely centroid-level analysis cannot show such aspects, due to the fact that relationships are always computed regarding the centroid of the group. Conversely, restricting the dataset to 1st (inter-specific) pullers could obscure important functional impacts (e.g., in centroid displacement) of same-species pulling events. While, we are, of course, interested in which species pulls another species more (inter-species level only), we also want to showcase to which degree each species moves individuals of the group the most (regardless of their species), functionally driving group movement. Furthermore, imposing such a strict limitation to analysis of the dataset would diminish statistical power to robustly measure differences across all dimensions that were studied (frequency, efficiencies, ratios, proportions, etc).

In conclusion, if the dyad-level analyses were removed from the manuscript or moved to the supplementary materials, all the aforementioned information would be lost or hidden to the reader (a high number of paragraphs of text would refer to SFigures, instead of the main ones). Nevertheless, we understand the reviewer's point, and fully agree that it is also fundamental to highlight interactions

between a focal individual and the centroid of the rest of the group, as it could provide different interpretations (although, in our case, the results are the same). Indeed, their valuable suggestion allows us to present a multiscale analysis of movement influence, both at group centroid and dyadic levels. To that end, we have altered Fig.2 to include these results and also have reshaped the text to give more emphasis to the centroid level analyses.

Social influence is hierarchically-distributed over multiscale dimensions representing role specializations: fish (particularly goatfish) drive environmental exploration, deciding where, while the octopus decides if, and when, the group moves (L33-36).

Lastly, to provide a multiscale overview of influence, we also quantified both pulling and anchoring frequencies between a given focal individual and the centroid of the group (see Methods for details and other complementary analyses). (L126-L129)

By analyzing pull-anchor dynamics and social influence at a group-level (i.e., not relatively to another individual, but relative to the group's centroid), we found a similar influence distribution, whereby blue goatfish are the main pullers (Tukey HSD, $p < 0.05$, Table S15, Fig. 2E), and octopuses are the main anchorers of the group's centroid (Tukey HSD, $p < 0.05$, Table S16, Fig. 2F). Thus, in general collective movement, group dynamics can be divided into two functional roles: goatfish, in particular blue goatfish, explore space and determine where the group may go, whereas the octopus decides if, and when, the group goes. (L187-193)

Our findings show how organisms phylogenetically separated for ~550M years can coordinate during collective hunting based on multidimensional and multiscale influence networks. (L371-372)

R1.2. Line 314 – 332: Here, data sets from two different octopus populations (Eilat and El Quseir) are compared. Please add to this paragraph one sentence clarifying to the reader that differences in web-over characteristics between solitary and group hunting octopuses may also be due to general behavioral differences between the two populations studied.

REPLY: We understand the confusion and apologize for not making the information clearer. Solitary hunting data was also acquired in Eilat, not only El Quseir, as specified in the Methods section (see also the full data structure regarding web-over characteristics in Data S5). As data from solitary hunting in Eilat was scarce, data from El Quseir was also included to enable more statistical robustness. As a precaution, before testing for differences between solitary and group hunting, we explicitly compared web-over frequency and duration during solitary hunting between the two locations. We found no significant differences. Thus, differences in web-over characteristics between solitary and group hunting octopuses are not significantly driven by different field locations. We previously had this information in the Statistics section of the Methods, but have moved it to the start of the Methods, immediately after we mentioned that additional data from El Quseir was used. We hope this streamlines the manuscript and makes the information clearer.

Together with an additional ~30 minutes dataset in Egypt (El Quseir 26.1014° N, 34.2803° E), video from the 'Zoom Camera' was used to quantify temporal characteristics of web-overs in solitary octopuses. As the least subjective component of octopus foraging, we registered web-over frequency and considered web-over duration as the timespan where the octopus exhibits whitening of the interbrachial web skin over a specific habitat feature. There were no significant differences between data collected in Israel and Egypt during solitary hunting, either regarding web-over frequency (Poisson GLMM, $n = 14$, $z = -0.661$, $p = 0.509$) or duration (Gaussian GLMM, $n = 107$, $z = -1.166$, $p = 0.244$). Videos were all filmed at 25 fps with 4k resolution, and cameras were synchronized in Adobe Premiere via the timestamp of an underwater horn at the start of all recordings. (L394-403)

R1.3. Line 444 – 447: I still don't understand why it was necessary to divide the hunting events into blocks of 100 seconds. Does that mean that a hunting event with a duration of more than 100 seconds was cut and considered as two separate events? When the authors are able to calculate the duration of hunting events ("The smallest duration of a hunting event was 100 seconds, ...") they must be able to define precise start and end points for such events. Wouldn't it be more correct and less confusing to clearly define hunting events and restrict analysis to these events rather than just chopping the whole footage into 100-second long chunks irrespective of when in these chunks a hunting event took place? How was the term "hunting event" actually defined? As a certain period of time before and after a web-over? How was the relationship between hunting events and pull/anchor events? Were only pull/anchor events analyzed that happened temporally close to web-overs, or were all pull/anchor events that happened within the 100-second block analyzed? If the latter was the case, and pull/anchor events were independent from web-overs, why fragmenting the footage at all? In addition, this paragraph still does not give a good overview of the actual sample size. The number of individuals given in line 449 – 450 is misleading because it is artificially enlarged by chopping up the footage into 100-second long chunks. I know it is hard to say from footage taken from wild animals, but I think it is important to know how many different octopus individuals were involved in this study to judge the generalizability of the results amongst *Octopus cyanea*. Otherwise, the limitation on generalizability of conclusions should be mentioned in the manuscript.

REPLY: We apologize for the lack of clarity regarding the nomenclature used, which we believe is the root of the misunderstanding. By 'hunting events', we meant scenes filmed with groups of octopus and fishes (13 hunting groups in total, as stated in L453). We have now changed all instances of 'hunting events' to 'hunting scenes' (mostly in the methods section, when mentioning filming or recording) or simply 'hunting groups', for clarity. In the Statistics section, we specify:

Thus, as a first step, we implemented auto-regressive model structures based on the subgroup number within each group, thus not only maintaining identity, but also weighing temporal auto-correlation between subgroups within the same group (i.e. subgroup + 0|group/individual). When this approach failed, i.e. the models failed to converge or were outperformed by simpler GLMMs, we used a nested structure of random effects with individual identity nested within groups, i.e. maintained across different subgroups (i.e. 1|group/individual/subgroup). For web-over temporal characteristics, we also included field site as the

first term of the nested random effects, given additional footage used from Egypt when the octopus was hunting alone. (L619-627)

To be clear, we considered that the 'start' of a hunting scene was when we found an octopus-fish group hunting (which of course may not be its 'real' start - but, in the case of the 100 s one, it was), and the 'end' of the hunting scene was when the group disbanded (or when camera batteries died). Thus, a hunting scene not necessarily connected to the occurrence of web-overs, punches, or pull-anchor movements, but to the association of the octopus with fish(es). If we had divided the scene around metrics such as before/after a web-over/pulling event, we would then be unable to quantify frequencies in standardized time units, or differences in frequencies or proportions driven by group composition. For instance, we would not be able to quantify that, for the same time length, there are more initiation attempts with blue goatfish in the group, than in groups where there are no blue goatfish. Parallely, it would not make sense to quantify web-overs if their occurrence was somehow used as criteria to determine the start/end of a hunt. For additional clarification, when, for example, one web-over (or initiation event) started in block 1 and finished in block 2, we always counted that occurrence as part of block 1.

Why was it then important to split hunting scenes and standardize group length to independently calculate metrics? One could assume that we are dealing with a linear system, and simply divide the total occurrence of metrics (let's say pull/anchors) by the duration of the hunting scene, obtaining the overall frequencies or rates. However, the explanation as to why that would be a mistake is reflected in Fig. S1. From the speed profiles of each individual, it is clear that the frequency of events in time is non-linear (a common feature of complex systems), especially in long duration hunting groups. To further illustrate our point, in the block between 0 and 100 s there is 1 pulling event. In the next block, between 100 and 200 s, there are 12 pulling events! (see 3.pull_anchor script). Thus, simply dividing the number of occurrences by different time lengths, assuming linearity, would be an error and provide false/oversimplified conclusions. For example, one could conclude that blue goatfish groups are ALWAYS more mobile than groups with blacktip groupers across time, whereas we now know that there is an inherent variability. Thus, had we not done the splitting, not only would the range of said variability within the system be hidden, but more importantly our analyses would not show the true range of the dynamics of the system. While we believe that the shortest group duration is a reasonable time length to be chosen as the basis for analyses, it is true that using this particular value could introduce some kind of artefacts into our data. To make sure that was not the case, we have now computed the main metrics (pulling and anchoring frequencies) using 200 and 300 s intervals as the standard time length. We found no differences in estimates compared to using 100 s (Fig S18, Tables S47-51).

Thus, the division in 100 s blocks (or into 'subgroups') allowed for more detailed comparisons among groups filmed in scenes with different lengths, also providing more statistical power while, of course, maintaining the hierarchical structure of the data present in the statistical models, regarding hunting group and individual identities (as seen in the text above, see also R scripts for specific model architectures). The numbers of individual identities per species (already considered in the statistical models) were also

present in Data S1, but we have now included them explicitly in the text in the same paragraph. We hope these changes make the methodology and the overall manuscript clearer:

In total, 3.5 h of collective hunting were reconstructed (example in Fig. 1B, and Video S2), composed of 13 different scenes representing different groups of interspecific hunting. From these 13 groups, we collected data and maintained the individual identity of 13 octopuses, 22 blacktips, 12 barbel goatfish, 10 lyretails, 20 blue goatfish, and 4 yellow goatfish. The size of the groups varied between 2 and 10 individuals, with an average presence of 5.72 individuals, always with 1 octopus present. The shortest scene recording of a hunting group was 100 seconds, while the largest reached approximately 1800 seconds (limited by camera battery). As movement frequency was non-linear across time, particularly in long duration groups (Fig. S1), we standardized length variability to better show the dynamics of the groups. We used the smallest scene length of a hunting group (100 seconds) to divide all hunting groups in 100 second blocks, totaling 107 hunting subgroups. In the end, our reconstructed subgroups provided a sample size of 132 blacktips, 95 barbel goatfish, 42 lyretails, 107 octopuses, 102 blue goatfish, and 23 yellow goatfish (see the Statistics section for techniques used to deal with data dependency). We further re-ran analyses on the main estimated parameters using 200 (Fig. S18A-B) and 300 (Fig. S18C-D) s blocks, and confirmed that the specific time length chosen did not impact our results. Details on number of blocks per group, and overlap of species in each subgroup/group is provided in Data S1, and dyadic sample sizes are shown in Table S51. (L453-470)

R1.4. Line 805 – 808: Please add to the figure caption the meaning of the red tracks in the middle panel of Fig. 1B and the color-coded tracks in the bottom panel of Fig. 1B.

REPLY: We have performed the requested changes:

We used a stereocamera rig (top) from which habitat features were identified (middle) and camera positions derived (in red) using structure-from-motion, enabling the calculation of relative 3D track positions (bottom). Colors show individuals present from different species, with large dots representing manual annotations and small dots showing smoothed interpolated tracks. (L822-826)

R1.5. Line 861 – 863: Please also add to the figure caption that here the behavior of two different populations of octopus are compared.

REPLY: Please see reply to **R1.2**.

R1.6. Fig. 2B: Error bars of octopus and yellow goatfish are truncated at the y-axis. The same applies to the blue goatfish in Fig. 2C.

REPLY: Indeed, error bars were truncated to not exceed graph limits. However, the standard error is the same, either being upper or lower bounded. As the information is already present in the graph, and other possible pairwise comparisons do not exist beyond the bounds, we opted to truncate the error bars, adding a dashed line indicating said truncation. We added this information to the Figure caption.

As upper and lower standard error estimates are similar, error bars are truncated to fit graph limits where applicable (dashed lines). (L849-850)

R1.7. Fig. 2C: Please add to the figure caption the meaning of the asterisk next to the octopus in Fig. 2C. If the asterisk relates to Fig. 2D, it should be removed, since Fig. 2D is not a cutout of Fig. 2C.

REPLY: The asterisk refers to changes in general influence, i.e., the purple dimension (the fusion between blue and red). We have explicitly added this information to the figure caption now. The purple asterisk and box specifically take the reader to Fig. 2D, where the purple dimension is explicitly plotted. Thus, while not a direct cut-out per se, we believe it helps visualize the link between the metrics analyzed in the two panels.

The purple asterisk refers to significant differences in general influence (calculated as the sum of pulling and anchoring frequencies), i.e., the purple dimension (the fusion between blue and red). (D) explicitly showcases the general functional influence that each species has in influencing the movement of others. (L838-840)

R1.8. Fig. 3B: Vertical error bar of barbel goatfish is truncated at the top.

REPLY: See reply to **R1.6**.

R1.9. Fig. 3D: In the bottom right, the small picture of blue goatfish is on the top and of the blacktip at the bottom for the first two box plots but changed for the last boxplot. The order of species should be consistent to avoid confusion.

REPLY: We thank the reviewer for their suggestion and have implemented the requested changes in Fig. 3 and 5.

R1.10. Fig. 4A and B: I still don't think that still images are useful when one aims at displaying dynamic events. There is sufficient space in this figure to replace the two still images by two series of images taken at different times throughout the specific partner control behavior. The size of the bar plot in panel D can be reduced by 50%.

REPLY: We added a series of images displaying the dynamic events, as suggested.

R1.11. Fig. 4C: The dashed box around the octopus is not necessary. Error bars are missing for all species. Please indicate in the axis labels which axis is represented by the green and yellow letters that indicate significant differences.

REPLY: We have removed the dashed box. Error bars cannot be calculated for each species as these are proportions (and statistical tests are performed in a series of pairwise comparisons). Apologies for our mistake, we have replaced 'normalized frequency' for proportion in the figure caption. Apologies for the color oversight as well, we have now added the information that executed displacements are in yellow, while received displacements are in green (also in the figure captions).

(C) shows the relative ranking of each species towards others in terms of aggressive interactions via received (green) and executed (yellow) displacements, using a similar methodology to Fig. 2.

Displacements are normalized by the number of groups/blocks where species are present, and are therefore given as a proportion. (L866-869)

R1.12. Fig. 4D: Please remove the dashed box. The y-axis label should read “Proportion of received punches”.

REPLY: Dashed box was removed and the label has been updated.

R1.13. Fig. 4E: Replace upper case by lower case letter “D” in the word displacement in the x-axis label.

REPLY: Done.

R1.14. Fig. 5C: What is meant by “Web-over frequency (per block of 100 s)”, the number of web-overs within 100 seconds? If so, it might be better to write “Number of web-overs in 100 s” or to calculate an actual frequency (events/second) value. As in Fig. 3D, it might help to keep the position of specific fish species constant (blue goatfish on top/blacktip at the bottom) in the pictogram next to the box plots in the upper right corner of the panel.

REPLY: We have changed to Number of web-overs in 100 s, as suggested.

Reviewer #2:

R2.1. Thank you to the authors for thoughtfully incorporating my feedback and making the necessary revisions. The paper now reads smoothly, greatly enhancing the clarity and accessibility of the science. I have a few additional minor suggestions outlined below. Overall, congratulations on an excellent study

REPLY: We thank the reviewer for their thoughtful comments, which have greatly improved the readability and (importantly) the accessibility of the science. There seems to have been a small problem with line numbering in the reviewer's version, but we believe we have found the instances referred.

Main text

R2.2. Line 317: I realise there are multiple fish species involved but I think for readability, keep the term as 'octopus-fish' rather than 'octopus-fishes', you also use the term 'octopus-fish' below so best to stay consistent.

REPLY: We performed the suggested change (L75).

R2.3. Line 319: For consistency, it would be worthwhile changing the term 'induces' to 'stimulates' as you've done above.

REPLY: We performed the suggested change (L77).

R2.4. Line 329: Consider replacing the term 'conjectural' with 'speculative'.

REPLY: We performed the suggested change (L87).

R2.5. Line 1978: remove the term 'predominately' or 'highly', you don't need both here.

REPLY: We have removed 'predominantly' (L315).

Reviewer #3:

R3.1. The authors have done a superb job at addressing my concerns (much more thoroughly than I had anticipated), and combined with their addressing of the other reviewer's comments, has allayed the few concerns I had about the veracity of their conclusions based on the analyses. Congratulations on a really exciting and novel paper.

REPLY: We thank the reviewer for their comments, which have improved our manuscript, approval of the methodologies used, and their final positive appreciation.

Decision Letter, second revision:

Our ref: NATECOLEVOL-23123085B

4th July 2024

Dear Dr. Sampaio,

Thank you for your patience as we've prepared the guidelines for final submission of your Nature Ecology & Evolution manuscript, "Multidimensional influence drives leadership and composition-dependent success in octopus-fish hunting groups" (NATECOLEVOL-23123085B). Please carefully follow the step-by-step instructions provided in the attached file, and add a response in each row of the table to indicate the changes that you have made. Please also check and comment on any additional marked-up edits we have proposed within the text. Ensuring that each point is addressed will help to ensure that your revised manuscript can be swiftly handed over to our production team.

****We would like to start working on your revised paper, with all of the requested files and forms, as soon as possible (preferably within two weeks). Please get in contact with us immediately if you anticipate it taking more than two weeks to submit these revised files.****

In recognition of the time and expertise our reviewers provide to Nature Ecology & Evolution's editorial process, we would like to formally acknowledge their contribution to the external peer review of your manuscript entitled "Multidimensional influence drives leadership and composition-dependent success in octopus-fish hunting groups". For those reviewers who give their assent, we will be publishing their names alongside the published article.

Nature Ecology & Evolution offers a Transparent Peer Review option for new original research manuscripts submitted after December 1st, 2019. As part of this initiative, we encourage our authors to support increased transparency into the peer review process by agreeing to have the reviewer comments, author rebuttal letters, and editorial decision letters published as a Supplementary item. When you submit your final files please clearly state in your cover letter whether or not you would like to participate in this initiative. Please note that failure to state your preference will result in delays in

50accepting your manuscript for publication.

Cover suggestions

We welcome submissions of artwork for consideration for our cover. For more information, please see our guide for cover artwork.

Nature Ecology & Evolution has now transitioned to a unified Rights Collection system which will allow our Author Services team to quickly and easily collect the rights and permissions required to publish your work. Approximately 10 days after your paper is formally accepted, you will receive an email in providing you with a link to complete the grant of rights. If your paper is eligible for Open Access, our Author Services team will also be in touch regarding any additional information that may be required to arrange payment for your article.

Please note that *Nature Ecology & Evolution* is a Transformative Journal (TJ). Authors may publish their research with us through the traditional subscription access route or make their paper immediately open access through payment of an article-processing charge (APC). Authors will not be required to make a final decision about access to their article until it has been accepted. Find out more about Transformative Journals

Authors may need to take specific actions to achieve compliance with funder and institutional open access mandates. If your research is supported by a funder that requires immediate open access (e.g. according to Plan S principles) then you should select the gold OA route, and we will direct you to the compliant route where possible. For authors selecting the subscription publication route, the journal's standard licensing terms will need to be accepted, including <https://www.nature.com/nature-portfolio/editorial-policies/self-archiving-and-license-to-publish>. Those licensing terms will supersede any other terms that the author or any third party may assert apply to any version of the manuscript.

Please use the following link for uploading these materials:
[REDACTED]

[REDACTED]

Reviewer #1:

Remarks to the Author:

I thank the authors for addressing my concerns thoroughly and for their detailed replies to my comments. Although I still think the main figures should be based on a dataset that does not include intraspecific interactions, the issue of bias due to such interactions should now be clear enough in the text to allow the readers to understand the limitations of the study. Congratulations on a very interesting paper!

Final Decision Letter:

25th July 2024

Dear Eduardo,

We are pleased to inform you that your Article entitled "Multidimensional social influence drives leadership and composition-dependent success in octopus-fish hunting groups", has now been accepted for publication in Nature Ecology & Evolution.

Over the next few weeks, your paper will be copyedited to ensure that it conforms to Nature Ecology and Evolution style. Once your paper is typeset, you will receive an email with a link to choose the appropriate publishing options for your paper and our Author Services team will be in touch regarding any additional information that may be required

Due to the importance of these deadlines, we ask you please us know now whether you will be difficult to contact over the next month. If this is the case, we ask you provide us with the contact information (email, phone and fax) of someone who will be able to check the proofs on your behalf, and who will be available to address any last-minute problems . Once your paper has been scheduled for online publication, the Nature press office will be in touch to confirm the details.

52Acceptance of your manuscript is conditional on all authors' agreement with our publication policies (see www.nature.com/authors/policies/index.html). In particular your manuscript must not be published elsewhere and there must be no announcement of the work to any media outlet until the publication date (the day on which it is uploaded onto our web site).

Please note that *Nature Ecology & Evolution* is a Transformative Journal (TJ). Authors may publish their research with us through the traditional subscription access route or make their paper immediately open access through payment of an article-processing charge (APC). Authors will not be required to make a final decision about access to their article until it has been accepted. Find out more about Transformative Journals

Authors may need to take specific actions to achieve compliance with funder and institutional open access mandates. If your research is supported by a funder that requires immediate open access (e.g. according to Plan S principles) then you should select the gold OA route, and we will direct you to the compliant route where possible. For authors selecting the subscription publication route, the journal's standard licensing terms will need to be accepted, including <https://www.nature.com/nature-portfolio/editorial-policies/self-archiving-and-license-to-publish>. Those licensing terms will supersede any other terms that the author or any third party may assert apply to any version of the manuscript.

We welcome the submission of potential cover material (including a short caption of around 40 words) related to your manuscript; suggestions should be sent to Nature Ecology & Evolution as electronic files (the image should be 300 dpi at 210 x 297 mm in either TIFF or JPEG format). Please note that such pictures should be selected more for their aesthetic appeal than for their scientific content, and that colour images work better than black and white or grayscale images. Please do not try to design a cover with the Nature Ecology & Evolution logo etc., and please do not submit composites of images related to

your work. I am sure you will understand that we cannot make any promise as to whether any of your suggestions might be selected for the cover of the journal.

You can generate the link yourself when you receive your article DOI by entering it here: <http://authors.springernature.com/share>.

[REDACTED]

P.S. Click on the following link if you would like to recommend Nature Ecology & Evolution to your librarian <http://www.nature.com/subscriptions/recommend.html#forms>

** Visit the Springer Nature Editorial and Publishing website at www.springernature.com/editorial-and-publishing-jobs for more information about our career opportunities. If you have any questions please click here.**